# Machine learning for observation bias correction with application to dust storm data assimilation

Jianbing Jin[1], Hai Xiang Lin[1], Arjo Segers[2], Yu Xie[1], and Arnold Heemink[1]

[1]Delft Institute of Applied Mathematics, Delft University of Technology, Delft, the Netherlands
[2]TNO, Department of Climate, Air and Sustainability, Utrecht, The Netherlands

**Correspondence:** Jianbing Jin (J.Jin-2@tudelft.nl)

**Abstract.** Data assimilation algorithms rely on a basic assumption of an unbiased observation error. However, the presence of inconsistent measurements with nontrivial biases or inseparable baselines is unavoidable in practice. Assimilation analysis might diverge from reality, since the data assimilation itself cannot distinguish whether the differences between model simulations and observations are due to the biased observations or model deficiencies. Unfortunately, modeling of observation biases or baselines which show strong spatiotemporal variability is a challenging task. In this study, we report how data-driven machine learning can be used to perform observation bias correction for data assimilation through a real application, which is the dust emission inversion using $PM_{10}$ observations.

$PM_{10}$ observations are considered as unbiased, however, a bias correction is necessary if they are used as a proxy for dust during dust storms since they actually represent a sum of dust particles and *non-dust* aerosols. Two observation bias correction methods have been designed in order to use $PM_{10}$ measurements as proxy for the dust storm loads under severe dust conditions. The first one is the conventional chemical transport (CTM) model that simulates life cycles of *non-dust* aerosols. The other one is the machine learning model that describes the relations between the regular $PM_{10}$ and other air quality measurement. The latter is trained by learning using two years of historical samples. The machine learning based *non-dust* model is shown to be in better agreements with observations compared to the CTM. The dust emission inversion tests have been performed, either through assimilating the raw measurements, or the bias-corrected dust observations using either the CTM or machine learning model. The emission field, surface dust concentration and forecast skill are evaluated. The worst case is when we directly assimilate the original observations. The forecasts driven by the posterior emission in this case even results in larger errors than the reference prediction. This shows the necessities of bias correction in data assimilation. The best results are obtained when using the machine learning model for bias correction, with the existing measurements used more precisely and the resulting forecasts close to reality.

## 1 Introduction

For centuries, East Asia experienced regular dust storms in the spring time. Those dust events mainly originated from the dust source regions of the Gobi and Taklamakan deserts. Annually, thousands tons of "yellow sands" are blown

eastward over the densely populated areas in China, the Korean peninsula, and Japan by the prevailing winds. Dust storms can also carry irritating spores, bacteria, viruses and persistent organic pollutants (WMO, 2017). Next to the human health, the resulting low visibility can cause a severe disruption of transportation systems. For example, more than 1,100 flights have been delayed/canceled in Beijing after the city was struck by a choking dust storm in early May 2017.

A large number of dust simulations models has been developed over the past decades (Wang et al., 2000; Gong et al., 2003; Liu et al., 2003). These chemical transport models help to understand the life cycles of the dust storms, and are also used for dust forecasts and to aid early warning systems. Apart from advances in simulation of dust storms, progress has also been made in the monitoring of dust or general aerosol loads. Field station networks are constructed to observe the in-situ particulate matter (PM) levels over densely populated regions (Li et al., 2017a). Ground-based sun photometers, e.g., the global Aerosol Robotic Network (AERONET) (Cesnulyte et al., 2014), are widely used to monitor column-integrated aerosol profiles. Satellite onboard instruments such as Moderate Resolution Imaging Spectroradiometer (MODIS) (Remer et al., 2005), Cloud-Aerosol Lidar and Infrared Pathfinder Satellite Observations (CALIPSO) (Sekiyama et al., 2010), and Advanced Himawari Imager/Himawari-8 (Yoshida et al., 2018) provide the measurements of airborne particles with further wide coverages. These measurements could be used to calibrate the parametrization in the dust simulation models and to evaluate their ability to forecast dust concentrations. Moreover, the observations could be combined with a dust modeling system through data assimilation to improve the forecast skills.

A wide variety of data assimilation techniques have been used with dust simulation models, including variational methods (Yumimoto et al., 2008; Niu et al., 2008; Gong and Zhang, 2008; Jin et al., 2018) and ensemble-based sequential methods (Lin et al., 2008; Sekiyama et al., 2010; Khade et al., 2013; Di Tomaso et al., 2017). In these systems, the available observations are either used to estimate the model states (dust concentrations) or to reduce uncertainties in the emissions and/or other model parameters. Challenges for dust assimilations include development of more and more accurate dust simulations, and use of new types of observations including vertical profiles from Lidars and latest satellite observations. A further challenge for any assimilation system is the proper definition of the observation and representation errors, as well as characterization of biases.

In general, the commonly used data assimilation schemes all rely on the basic assumption of an unbiased observation. In real applications, however, measurement biases are often unavoidable. In the presence of biases, it is impossible to determine whether a difference between an *a prior* simulation and an observation are due to the biased observations or model deficiencies. The biases might lead to assimilations that diverge from reality (Lorente-Plazas and Hacker, 2017). A well known example of observation biases is in radiance observation assimilation systems in presence of clouds (Eyre, 2016; Berry and Harlim, 2017). To avoid problems with these biases, up to 99% of cloudy observed measurements are discarded although they may also contain valuable information. If dust storms are coincident with clouds, it is also possible that in satellite retrieval algorithms clouds are mistaken for dust, leading to strong biases in the data to be assimilated (Jin et al., 2019).

Another example where observation biases are important is when ground-based $PM_{10}$ measurements are assimilated in dust simulation models. Due to the high temporal resolutions and the rather dense observation network, the ground-based air quality observing network has become a powerful source of measurements on dust aerosols. The records, mainly the $PM_{10}$ feature, were widely used to calibrate, assess or estimate the dust model (Lin et al., 2008; Wang et al., 2008; Huneeus et al., 2011; Yumimoto et al., 2016; Benedetti et al., 2018). However, the observed $PM_{10}$ concentrations do not only consist of dusts, but are actually the sum of the dust and other regular particles. The latter one are emitted not only from anthropogenic activities such as industries, vehicles, and households, but also from natural sources such as wild fires and sea spray. In this paper we will simply refer to these particles as the *non-dust* fraction of the total $PM_{10}$. The concentrations of *non-dust* aerosols in urbanized areas could be substantial, reaching values up to 500 $\mu$g/m$^3$ (Shao et al., 2018).

Although $PM_{10}$ observations include a nontrivial bias, the wide spread availability makes them still useful in dust storm assimilation system. During dust storm events, extreme high peaks of more than 1000-2000 $\mu$g/m$^3$ $PM_{10}$ are recorded which can be attributed mainly to dust. If these would be assimilated directly in dust simulation model, ignoring the fact that at least some part represents *non-dust*, the assimilation system would diverge to states that overestimate the dust load. In case of less severe dust events, the dust analysis divergence would then become extremely critical.

However, modeling of observation biases is very challenging when they have strong spatial and temporal variabilities. Little progress has been made in bias correction of full-aerosol measurements for their use in dust storm data assimilation. Lin et al. (2008) selected only $PM_{10}$ observations for assimilation when at least one occurrence of dust clouds was reported by the local stations. In Jin et al. (2018), it was found that on sites with both $PM_{10}$ and $PM_{2.5}$ observations, only the $PM_{10}$ concentration increased during a dust episode, while the $PM_{2.5}$ concentrations were not affected and remained at a constant level. Besides, Xu et al. (2017) and Jin et al. (2018) suggested a strong correlation between $PM_{2.5}$ and *non-dust* $PM_{10}$. Therefore, a very simple *non-dust* $PM_{10}$ baseline removal (called observation bias correction) was proposed, in which the available $PM_{2.5}$ was used to approximate the *non-dust* $PM_{10}$ (or baseline) during a dust event by:

$$PM_{10}^{non\text{-}dust} \; = \; b \; + \; r \times PM_{2.5} \tag{1}$$

where the $b$ and $r > 1$ are linear regression parameters based on a 24-hour history of measurements before arrival of the dust storm. The aforementioned methods either exclude a selection of the measurements, which may still contain useful information, or work under ideal circumstance only when a simple correlation $\mathcal{R}$ between $PM_{10}$ and $PM_{2.5}$ is valid. For instance, in the dust event studied in Jin et al. (2018) the application of Eq.1 in many sites failed since $R$ is weak. To have a quality-assured bias correction, Eq.1 is performed only when the *Pearson* correlation coefficient $\mathcal{R} > 0.8$. Consequently, measurements in around 45% sites are rejected in that case. To fully exploit the dust information present in total PM observations, a more advanced method is needed. In this paper we proposed two methods, either using a conventional chemistry transport model, or a machine learning model.

A chemistry transport model (CTM) implements all available knowledge on emission, transport, deposition, and other physical processes in order to simulate concentrations of trace gases and, important here, aerosols. Daily air quality forecasts are often provided using such CTMs. A simulation model for dust storm events is usually just a CTM with all tracers removed except dust; by using the full CTM, an estimate of the *non-dust* part of the aerosol load could be made. In this study, the LOTOS-EUROS CTM is used to simulate the dust as well as the *non-dust* aerosol concentrations. If the non-dust model was perfect, the difference between simulation and observed $PM_{10}$ would be unbiased, and assimilation could be applied to the combined dust and *non-dust* concentrations. In case of a dust storm event, it remains necessary to distinguish between the dust and *non-dust* part of the simulations since the two parts will have very different error characteristics. The dust part is quickly varying and has a large uncertainty, while the *non-dust* part is more smooth but very persistent in time and has a relatively small uncertainty. An assimilation system on the combined simulations should be able to handle these differences. However, the error attribution to their proper sources (dust and *non-dust* error) then becomes extremely critical as explained in Section 2.4. Since this paper focuses on dust during a severe event only, we will not explore the error characteristics of the *non-dust* part of the model. Therefore we will not apply an assimilation on the combined aerosol (dust and *non-dust*) model. Instead, the *non-dust* simulations will solely be used to remove the *non-dust* baseline from $PM_{10}$ observations.

Similar to the air quality forecast, the accuracy of a CTM for *non-dust* aerosols is hampered by lack of accurate input data. For example, the timely update of anthropogenic emission inventories is always a key issue for air quality forecasts. With the ever-increasing complexity and resolution, the CTMs are now becoming highly nonlinear and time-consuming. However, they may still not be able to identify explicit representations of the non-dust aerosol dynamics, especially regarding fine-scale processes.

In addition to the conventional CTM, we propose a new method for removing the *non-dust* part of the $PM_{10}$ observations which is based on machine learning (ML). Data-driven methods have already been proved to be a powerful tool to provide air quality forecasts for horizons of a few days, (e.g., Li et al. (2016); Fan et al. (2017); Li et al. (2017b); Chen et al. (2018)). Different from the chemical transport models which simulate the aerosol physical processes, machine learning models describe mathematical relations of input-output and trained by learning a large number of samples from historical records. Our machine learning system used a neural network, namely *long short term memory* (LSTM). The input is formed by air quality indices for a number of relevant tracers (PM2.5, $SO_2$, $NO_2$, CO, and $O_3$), as well as meteorology data. The output of the system is an estimate of the *non-dust* $PM_{10}$ concentration. The input features are to a large extent independent of the dust storms, even the $PM_{2.5}$ concentrations as shown in Jin et al. (2018); observations of $PM_{10}$ are excluded since excessive dust loads are visible mainly in this component. Recent development and the availability of open source machine learning tools provide a good opportunity to estimate the air quality indices using a data-driven machine learning models.

Whereas these are previous studies on dust storm data assimilation using various kinds of combined aerosol measurements, we are the first to investigate the necessities of bias correction for these full-aerosol observations in order to use them as 'real' dust measurements in a dust storm assimilation system. The adding values of observation

bias correction in dust emission inversion is explored through the ground-based $PM_{10}$ measurement assimilation. It can easily be applied to others general applications, e.g., remote sensing data assimilation. Our contributions are threefold. Firstly, we present and examine the conventional CTM for removing the *non-dust* part from $PM_{10}$ observations. Secondly, we design and examine a novel machine learning based bias correction which is data-driven and free of the time-consuming numerical CTMs. Thirdly, we evaluate the two *non-dust* aerosol model simulations by comparing to the $PM_{10}$ measurements during regular periods (rare dust events involved); we evaluate dust emission fields, surface dust concentration simulation and forecast skills which are obtained by either assimilating the raw $PM_{10}$ data, or bias-corrected measurements either using the CTM or machine learning model.

The paper is organized as follows. A brief description of our dust simulation model (LOTOS-EUROS/Dust) and the four dimensional variational data assimilation method for emission inversion are presented in Section 2. The biased observation representing error and its influence on the assimilation system are also explained. The two bias correction methods, the *non-dust* aerosol regional chemical transport model and a machine learning model, are discussed and the bias simulation is evaluated in Section 3. Section 4 reports the assimilation results using the two bias correction methods, and evaluates the forecast skills using independent measurements. Section 5 discusses the necessities of observation bias correction in assimilation works, highlights our key contributions.

## 2    Dust storm data assimilation system

### 2.1    Dust model

The dust storm event studied in this paper took place in East Asia in April 2015, and has already been used as a test case for assimilation experiments in Jin et al. (2018). The LOTOS-EUROS/Dust simulation model is used with similar configurations to our previous studies, which is configured on a domain from 15°N to 50°N and 70°E to 140°E, but with a higher model resolution of 0.25°. The model is driven by European Center for Medium-Ranged Weather Forecast (ECMWF) operational forecasts for horizons of 3-12 hours. The dust load is described by 5 aerosol bins within a diameter range 0.01 $\mu$m $< D_p <$ 10 $\mu$m. Physical processes included are emission, advection, diffusion, dry and wet deposition, and sedimentation. The dust emission scheme implemented in LOTOS-EUROS is mainly based on the formulation of horizontal saltation flux (Marticorena and Bergametti, 1995) and sandblasting efficiency (Shao et al., 1996). A terrain preference parameter $F_{ps}$ was used in the dust emission in Jin et al. (2018). This geographic dependent parameter was first introduced by Ginoux et al. (2001), and used to approximate the probability of having accumulated sediments that can be resuspended. In this work, $F_{ps}$ is disabled since the preference factor was found to limit the emission rate in some regions where the fine-scale topographic feature is actually unknown. Snapshots of a reference simulation of the dust episode has been performed and is shown in Fig.8(a).

## 2.2 Observation network

The China Ministry of Environmental Protection (MEP) has commenced to release the hourly-average measurements of atmospheric constituents including $PM_{2.5}$, $PM_{10}$, $CO$, $O_3$, $NO_2$ and $SO_2$ since 2013. A huge number of ground stations measuring these air quality indices have been established in densely populated areas. At the present, the monitoring network has grown to 1,500 field stations covering all over China as shown in Fig.1.

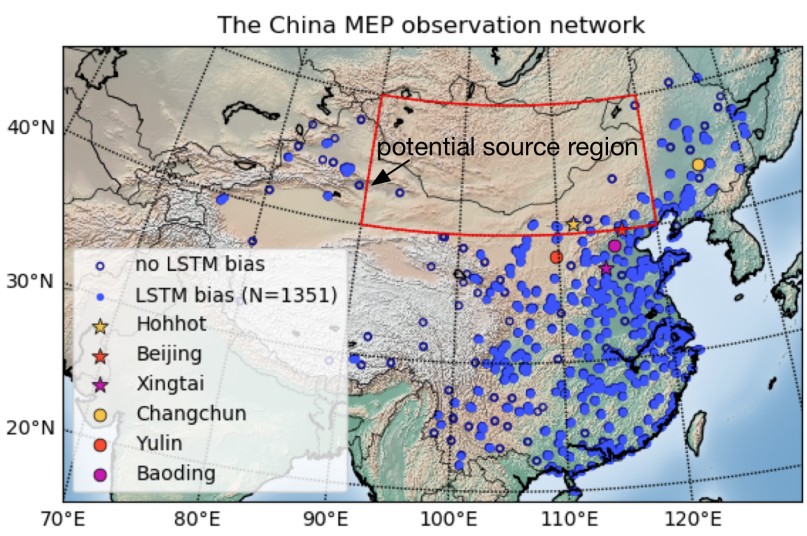

**Figure 1.** The China MEP air quality monitoring network and the potential dust storm source region. LSTM based *non-dust* $PM_{10}$ forecast are performed only in stations of blue dot (N=1351), while ones of black circles are skipped.

## 2.3 Reduced tangent linearization 4DVar

The assimilation system, which will be used to combine bias-corrected $PM_{10}$ observations with simulations, is based on a reduced-tangent-linearization four dimensional variational (4DVar) data assimilation. The goal of a 4DVar technique is to find the maximum likelihood estimation of a state vector, which is here the dust emission field $\boldsymbol{f}$, given the available observations over a time window. A common approach is to use an incremental formulation, which aims to find the optimal emission deviation $\delta\boldsymbol{f}$ as the minimum of the cost function:

$$J(\delta\boldsymbol{f}) = \frac{1}{2}\,\delta\boldsymbol{f}\,\mathbf{B}^{-1}\,\delta\boldsymbol{f} + \frac{1}{2}\sum_{i=1}^{k}(\mathbf{H}_i\mathbf{M}_i\delta\boldsymbol{f}+\boldsymbol{d}_i)^T\mathbf{R}_i^{-1}(\mathbf{H}_i\mathbf{M}_i\delta\boldsymbol{f}+\boldsymbol{d}_i) \tag{2}$$

where $k$ is the number of time steps within the assimilation window. The vector $\delta\boldsymbol{f}$ denotes a perturbation of the emissions with respect to the background one. For an observation time $i$, the innovation vector (length $m_i$) is defined

as the difference between the simulations and observations:

$$\boldsymbol{d}_i = \mathcal{H}_i(\ \mathcal{M}_i(\boldsymbol{f})\ )\ -\ \boldsymbol{y}_i \tag{3}$$

where $\mathcal{M}_i$ denotes the LOTOS-EUROS/Dust transport model, $\mathcal{H}_i$ is the operator that converts state variables into observation space, and $\boldsymbol{y}_i$ is the vector with dust observations at this time step $i$. The operators $\mathbf{H}_i$ and $\mathbf{M}_i$ denote linearizations of $\mathcal{H}_i$ and $\mathcal{M}_i$ around the reference emission vector $\boldsymbol{f}_b$. Following Jin et al. (2018), the errors in dust emission field were assumed to be only caused by the uncertainty in the friction velocity threshold in the dust windblown parametrization, and similar assumptions on the uncertainty are used to build an emission error covariance $\mathbf{B}$. The friction velocity threshold is perturbed with a spatially varying multiplicative factor $\beta$. $\beta$ is configured with a mean of 1 and a standard deviation of 0.1. In addition, an exponential profile of distance-based spatial correlation is posed on $\beta$s (Jin et al., 2018). The observation error term is weighted by an observation error covariance $\mathbf{R}$, for which the individual elements will be described in Section 4.1.

To reduce the computational cost in calculating the *tangent linear model* $\mathbf{M}_i$, a reduced-tangent-linearized 4DVar (Jin et al., 2018, 2019) is used. The simplified method is based on proper orthogonal decomposition (POD) of the background covariance $\mathbf{B}$ which efficiently carries out model reduction by identifying the few most energetic modes:

$$\mathbf{B}\ =\ \mathbf{U}\mathbf{U}^T\ \approx\ \tilde{\mathbf{U}}\tilde{\mathbf{U}}^T \tag{4}$$

$$\delta\boldsymbol{f}\ \approx\ \tilde{\mathbf{U}}\ \delta\boldsymbol{w}$$

where $\mathbf{U} \in \mathbf{R}^{P \times P}$ is the background emission covariance square root, with $P$ the size of the emission field of $O(10^4)$ elements. while $\tilde{\mathbf{U}} \in \mathbf{R}^{P \times p}$ is the truncation of $\mathbf{U}$ based on POD, with $p$ the reduced rank size of $O(10^2)$. The vector $\delta\boldsymbol{w} \in \mathbf{R}^p$ stores the transformed control variables.

The cost function of the reduced-tangent-linearization 4DVar is formulated as:

$$J(\delta\boldsymbol{w})\ =\ \frac{1}{2}\delta\boldsymbol{w}^T\delta\boldsymbol{w}\ +\ \frac{1}{2}\sum_{i=1}^{k}\left(\mathbf{H}_i\tilde{\mathbf{M}}_i\tilde{\mathbf{U}}\delta\boldsymbol{w}+\boldsymbol{d}_i\right)^T\mathbf{R}_i^{-1}\left(\mathbf{H}_i\tilde{\mathbf{M}}_i\tilde{\mathbf{U}}\delta\boldsymbol{w}+\boldsymbol{d}_i\right) \tag{5}$$

where $\tilde{\mathbf{M}}_i$ denotes the reduced tangent linear model with a rank $p$, which is approximated using the perturbation method. More details about the reduced-tangent-linearization 4DVar algorithm can be found in Jin et al. (2018).

## 2.4   Biased observation representing error

In real applications, the observations inevitably have biases which cannot be attributed to the model simulation, as following:

$$\boldsymbol{y}_i = \mathcal{H}_i(\ \mathcal{M}_i(\boldsymbol{f})\ )\ +\ \boldsymbol{b}_i\ +\ \boldsymbol{\sigma}_i \tag{6}$$

where $\boldsymbol{\sigma}_i$ is the vector of Gaussian distributed observation errors which have zero means and a known covariance matrix $\mathbf{R}_i$, and $\boldsymbol{b}_i$ denotes the vector of observation bias. In our application, the vector $\boldsymbol{y}_i$ contains the observed

$PM_{10}$ concentrations, while the aerosols released in the local anthropogenic activities and other *non-dust* related processes are referred as $\boldsymbol{b}_i$. Note that the $PM_{10}$ measurements themselves might also contain 'native' biases due to the incorrect sensor reading or systematic errors. However, this part of the bias in the $PM_{10}$ observations is unknown and not considered in this study.

In the course of data assimilation, it is impossible to determine whether the departures ($\boldsymbol{d}_i$) of the prior simulations from the observations are due to the biased observations $\boldsymbol{b}_i$ or emission errors $\delta\boldsymbol{f}$. Thus, the assimilation result will diverge from the true state when a bias is present. In complex dynamic models as the atmospheric transport model, the biases (*non-dust* aerosols) could have high spatial and temporal variabilities and is therefore difficult to quantify.

In this work, we proposed two methods to quantify the bias levels for the observation bias correction. The first one is the *non-dust* parts of LOTOS-EUROS chemical transport model (CTM) which simulates the aerosol life cycles including emission, transport and deposition. The second method is to describe the *non-dust* aerosol levels using a data-driven machine machine model. Details of these two methods are illustrated in Section 3.

In fact, both LOTOS-EUROS CTM and machine learning model are imperfect, and some biases might still exist after the correction. The former one is known to be limited by errors in the emission inventories, meteorological forecasts and all kinds of input sources. The latter is then hampered by the deficiency of the type model (e.g., insufficient to represent the complexity of the phenomenon), inadequate amount of training data. However, by combining the bias-corrected observation with the dust model, the assimilation will adapt to posteriors which are more close to reality.

There were a few studies that addressed both the model deficiency and uncertainty in observation bias simultaneously using either variational data assimilation (Dee and Uppala, 2009) or sequential filters (Dee, 2005; Lorente-Plazas and Hacker, 2017). Those assimilation schemes not only require a formulation of a model for the bias, but also need a quality-assured reference to describe the uncertainty of the bias model. The need to attribute errors to their proper sources is obviously a key part in any assimilation systems, but becomes especially critical when it involves bias correction. This is because a wrong error attribution will force the assimilation to be consistent with a biased source. If the source of a known bias is uncertain, assimilation without considering the uncertainty of bias model is the safest option (Dee, 2005). Therefore, these two *non-dust* models are solely set as references for the bias, and the uncertainties are not explored here.

### 2.5   Assimilation Window

Fig.2 shows a time line for the assimilation experiment around the April 2015 dust event, which is very similar to what was used in Jin et al. (2018). The dust event has a short duration, and therefore only a single assimilation window with a length of 36 hours is used. The dust emissions take place at the start of the window, while the observations become available at the end of the window since they are located downwind from the source region (see Fig.1). A long assimilation window is therefore necessary in order to estimate the correct emission parameters given the observations.

When we perform the assimilation analysis at April 15, 19:00, only the dust observations from April 15, 08:00 to 19:00 will be assimilated and they are calculated by subtracting the *non-dust* part (CTM based or ML based) from the $PM_{10}$ observations. After the analysis, the simulation model is used to perform a dust forecast for the next 12 hours using the newly-estimated emission parameters. A full-aerosol $PM_{10}$ forecast will then be calculated by adding the dust forecast and *non-dust* aerosol forecast, where the later again originates from either the CTM and machine learning model.

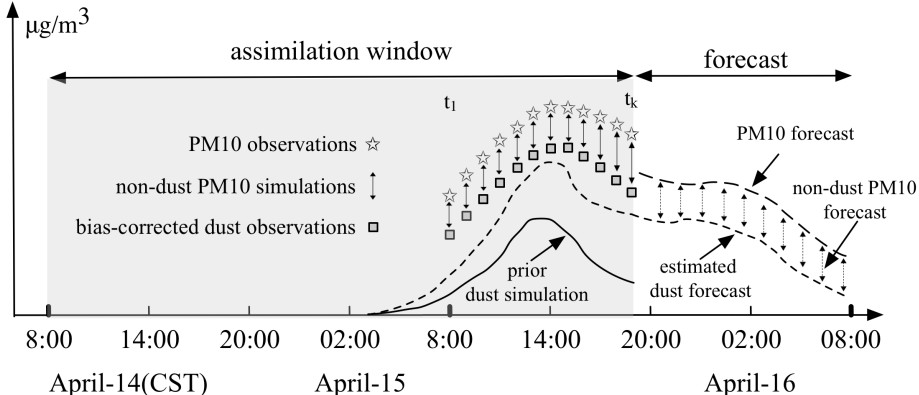

**Figure 2.** Timeline of observation availability, assimilation cycles and forecasts

## 3  Observation bias correction methods

Two systems are introduced to correct the *non-dust* bias when using $PM_{10}$ observations in a dust assimilation. The first one is CTM LOTOS-EUROS/*non-dust* model that simulates the physical processes of the *non-dust* aerosols. The latter is the machine learning model that estimates the *non-dust* aerosol based on historical records. The following sections describe the two methods in more detail.

### 3.1  Chemistry transport model (LOTOS-EUROS/*non-dust*)

The regional CTM LOTOS-EUROS/*non-dust* is configured similar to the LOTOS-EUROS/Dust used in the assimilation, but now includes all trace gases and *non-dust* aerosols. The configuration is similar to what is used for daily air quality simulations over China as described in (Timmermans et al., 2017). Anthropogenic emissions are taken from the Multi-resolution Emission Inventory for China (MEIC) inventory (*http://www.meicmodel.org*). Natural emissions included are the sea salts that are calculated online, biogenic emissions that are calculated online using the MEGAN model (Guenther et al., 2006), and wild fires which were taken from the operational GRAS product

(Kaiser et al., 2012). The LOTOS-EUROS full aerosol operational forecast over this modeling domain are released via the MarcoPolo-Panda projects through *(www.marcopolo-panda.eu)*.

The operational CTM Lotos-Euros over China is in its early phase of development as well as the other six CTMs used in the MarcoPolo-Panda project. The purpose of that project is to diagnose statistical differences between the ensemble model simulations and observations. An important objective is to determine ways by which the models can be improved. These differences are mostly attributed to inaccuracy in the weather forecast and errors in the adopted surface emissions (Brasseur et al., 2019; Petersen et al., 2019). Indeed, there is room for minimizing the forecast-observation differences using nudging methods like data assimilation, which requires considerable efforts and not yet exploited in that study.

## 3.2   Machine learning for *non-dust* $PM_{10}$ simulation

Given a set of training data, a machine learning algorithm attempts to find the relation between input and output. When a proper model is used, the machine learning algorithm can lean to reproduce the complex behaviors of a dynamic system. The description is purely based on the data, physical knowledge is not included. Machine learning algorithms are popular tools to forecast the air quality indices using the history records (Li et al., 2016; Fan et al., 2017; Chen et al., 2018; Lin et al., 2019). In this study, the machine learning algorithm used is the *long short term memory* (LSTM) neural network, which has demonstrated its ability in predicting time series problems (Li et al., 2017b).

The LSTM operator $\mathcal{L}$, which is configured with parameters $\boldsymbol{\theta}$, for predicting *non-dust* $PM_{10}$ can be described as:

$$\boldsymbol{b}^{t_0+t} = \mathcal{L}_{\boldsymbol{\theta}}(\boldsymbol{x}^{t_0},\ \boldsymbol{x}^{t_0-1},\ ...,\ \boldsymbol{x}^{t_0-m+1}) \tag{7}$$

where $\boldsymbol{b}^{t_0+t}$ represents the predictor, which is in this study the *non-dust* $PM_{10}$ concentration forecast $t$ hours in advance. The temporal correlation between the input and output features declines when $t$ increases. In our system, the maximum forecast period $t$ is 12 hours. The input vectors $\boldsymbol{x}^{t_0}, \boldsymbol{x}^{t_0-1}, ..., \boldsymbol{x}^{t_0-m+1}$ are the observed data of the past $m$ hours, which is set as 18 hours empirically. The input vectors consist of:

- hourly observations of $PM_{2.5}$, $SO_2$, $NO_2$, $O_3$, and CO from the ground based air quality network described in Section 2.2;

- observations of $PM_{2.5}$ at the nearby sites;

- local meteorological data (temperature and dew point at 2 m, wind speed at 10 m) which are taken from the LOTOS-EUROS model input and originate from the European Center for Medium-Ranged Weather Forecast (ECMWF).

The LSTM neural network parameters $\boldsymbol{\theta}$ are determined by minimizing the objective function $J_{\boldsymbol{\theta}}$ that represents the mean squared error of predictors $\boldsymbol{b}$ with respect to the measured values $\boldsymbol{y}^b$:

$$J_\theta = \frac{1}{m} \sum_{i=1}^{m} (\boldsymbol{b}_i - \boldsymbol{y}_i^b)^2 \tag{8}$$

The training dataset covers the period from January 2013 to March 2015. In other words, the LSTM model $\mathcal{L}$ is trained to best fit the samples from this period. The two months April and May 2015 in which the studied dust event occurred is set as the testing period.

Dust storms themselves occur with very low frequency. To our knowledge, the studied dust event is the most severe one since 2002, and there are no such large-scale dust events recorded in our training period. Note that cities that are close to the Gobi and Mongolia deserts might have experienced several small-scale dust events with limited increase of dust concentrations. However, the machine learning tries to find the global best fits for the whole training dataset. The default learning rate, which determines the weights are updated during training, on a simple sample is $10^{-4}$ in our machine learning algorithm. Therefore, the $PM_{10}$ records $\boldsymbol{y}^b$ are very close to the *non-dust* $PM_{10}$ concentrations, and the rare dust event records are not excluded from the training dataset for convenience and for the expected little impact on the training result. The regression model $\mathcal{L}$ is thus assumed to reflect only the relation between input features and the *non-dust* $PM_{10}$.

Note that including $PM_{10}$ observations in the series of input vectors will certainly improve the skill of the machine learning forecasts. However, the LSTM model would then lack the ability to discriminate between the dust and *non-dust* fractions in $PM_{10}$ during a dust event. Earlier studies showed that the input variables, including $PM_{2.5}$, are independent on the dust storm as illustrated in Jin et al. (2018).

For the non-dust $PM_{10}$ machine learning forecasts in a given site, observations from its nearby sites are also vital and are used in two ways. First, missing data records are unavoidable in an air quality monitoring network, while the LSTM model training requires an uninterrupted time series of features. In this study, data interpolations of air quality measurements ($PM_{10}$, $PM_{2.5}$, $SO_2$, $NO_2$, $O_3$ and $CO$) are performed using both a linear interpolation and a k-Nearest-Neighbor algorithm (Zhang, 2012) if a site has no more than 30% of missing data. Otherwise, all the measurements in the given sites are abandoned. Generally, more information available from the nearby sites will result in a more accurate interpolation. Second, learning in the presence of data errors is pervasive in machine learning, and the measurements from nearby stations are used to limit their influence. Data errors occur due to incorrect sensor readings, software bugs in the data processing pipeline, or even the inaccurate data interpolation. Statistical analysis tests have been conducted which did not only indicate a strong correlation between the non-dust $PM_{10}$ and air quality measurements in the given sites, but also show that the predictor (non-dust $PM_{10}$) is correlated to the observation indices (especially the $PM_{2.5}$) at its nearby sites. In order to eliminating errors caused by incorrect inputs at the modeling site, the measurements at the nearby stations are considered as the essential indices. In this study, a data instance will only be selected for training the LSTM model if there is at least one nearby site within an empirical radius 0.8°(approx 80 km), and a maximum of 3 nearby sites will be randomly selected where observation

20     stations are densely distributed. To save the computation costs on machine learning model training, only the $PM_{2.5}$ from the nearby sites are included as one of the inputs in this study.

      The machine learning model for non-dust $PM_{10}$ forecast is trained site by site, with the hyper-parameters shown in Table 1. With the following hyper-parameters, the machine learning model training takes several minutes for each site. The training in each site is independent, hence, the whole workload is highly parallelizable.

25 **Table 1.** LSTM hyper-parameters.

| LSTM layers | neurons per layer | epochs | batch size | forecast length (hours) |
|---|---|---|---|---|
| 2 | 30 | 50 | 64 | 0 or 12 |

      Fig.1 presents the original field observation network (N≈1500) established by the China Ministry of Environmental Protection (MEP) up to 2018, as well as the sites (N=1351) where LSTM based *non-dust* forecasts are performed. It is clear that the LSTM forecast cannot be performed in each monitoring site. A part of the sites is skipped due 30 to the lack of nearby sites, the rest are caused by high data missing rate in the training period.

### 3.3   Evaluation of *non-dust* $PM_{10}$ bias corrections

Our two bias models, LOTOS-EUROS/*non-dust* and LSTM, could both be used for air quality forecast operationally when there is no dust storm. Once a dust storm is observed, the dust emission inversion system will be enabled, the two non-dust $PM_{10}$ models will then be used in dust observation bias correction. The forecasts are expected to have 5 a good performance when dust is not present, and to underestimate the $PM_{10}$ levels in case of dust storms.

      Both the CTM LOTOS-EUROS and LSTM are tested to forecast *non-dust* $PM_{10}$ over April-May 2015. This period includes the 2 to 3 days dust event that is used as test case for the assimilation. Fig.3(a)~(c) show density plots comparing $PM_{10}$ observations with either LOTOS-EUROS/*non-dust* forecasts, or with LSTM forecast 0 hour and 12 hours in advance.

10       The CTM LOTOS-EUROS/*non-dust* in general underestimates the *non-dust* $PM_{10}$. The forecast results in a relatively large root mean square error (RMSE) 89.4 $\mu g/m^3$. This could be explained from the fact not all types of particulate matters, such as secondary organic aerosols, are included in the model, and some aerosol emissions are very difficult to estimate (e.g., wood burning by households). The two LSTM forecasts show on average a good agreement with the observations. The RMSEs of the forecasts by the two machine learning models in the two years 15 of training period are reduced to 55.9 and 60.7 $\mu g/m^3$, and in the two months of test period (excluding the dust event from April 14 to 16) they also stay at comparable low levels of 58.6 and 60.2 $\mu g/m^3$. As expected, a smaller forecast period $t=0$ hour gives a better result than the forecast over 12 hours.

      The scatters in the dust period (April 14 to 16) are denoted using different markers in Fig.3. The underestimation of $PM_{10}$ during the dust period (April 14 to 16) is visible in the bottom right corners of these plots.

20       When we perform the assimilation analysis at April 15, 19:00, the short period of $t=0$ hour forecast will be treated as the *non-dust* levels in the bias correction of the original $PM_{10}$ measurements. Note that here $t=0$ forecasts denote

the forecasts valid at each specific snapshot of the observations, while the 12 hours forecasts are valid 12 hours in advance, e.g., the non-dust $PM_{10}$ forecast (12 h) at April 16 07:00 is valid at April 15 19:00. Subsequently, the bias-corrected data are used to estimate the dust emissions over the past 36-hour window. Obviously, one important aim of the assimilation is to make a better forecast, in this study, the forecast skills will be evaluated in the following 12 hours from April 15, 19:00. Besides, the forecast is assessed by comparing the combined $PM_{10}$ forecasts to $PM_{10}$ observations. The LSTM forecast with $t=12$ hours in advance will be added to the dust storm forecast to build the combined aerosol forecast.

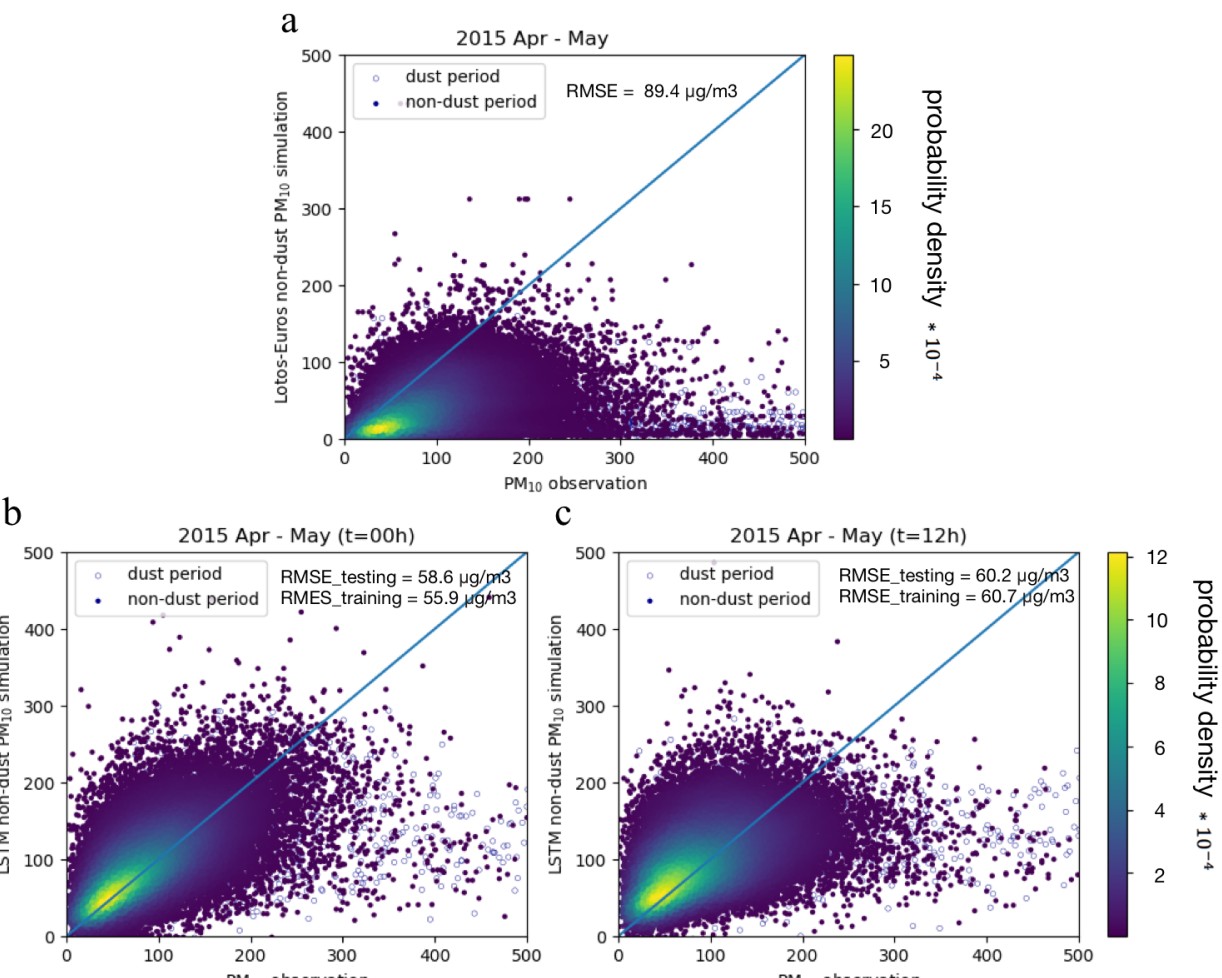

**Figure 3.** *Non-dust* $PM_{10}$ simulation evaluations. (a): LOTOS-EUROS/*non-dust* forecast vs. $PM_{10}$ measurements; (b): LSTM forecast 0 hour in advance vs. $PM_{10}$ measurements; (c): LSTM forecast 12 hour in advance vs. $PM_{10}$ measurements; (NOTE: the solid circles show the 5% random samples over the non-dust period from April to May 2015 while the hollow ones denote the 5% random ones from the dust period (April 14 to 16).

### 3.3.1 Spatial patterns at observation sites

30    To assess our two *non-dust* $PM_{10}$ models, Fig.4 shows the snapshots of the $PM_{10}$ measurements, LOTOS-EUROS/*non-dust* simulations, LSTM forecasts, and the corresponding bias-corrected dust observations at three timestamps: April 15 08:00, 19:00 and 22:00. These first two moments are the start and end of the observation interval in the assimilation window (only observations from the last 12 hours of the assimilation window are assimilated as shown in Fig.2), and observations at 22:00 is treated as independent data for cross-validation. At 08:00, actually only few stations close to the dust source area have already observed the dust storm. Some of the sites in central China observed high $PM_{10}$ concentrations which are believed to be caused by presence of *non-dust* aerosols. Nearly all the stations in north China reported this dust storm at 19:00 and 22:00, as a band covering central and northeast China, see Fig.4

(a.2)~(a.3). Fig.4 (b.1)~(b.3) shows that the LOTOS-EUROS/*non-dust* model forecasts quite stable and constant *non-dust* $PM_{10}$ levels, most of the simulated values are less than 100 $\mu g/m^3$. Subsequently, the corresponding bias-corrected dust measurements (see Fig.4 (c.1)~(c.3)) are very similar to the original $PM_{10}$ observations. This could be problematic when trying to measure the dust storm from the $PM_{10}$ observations; for instance at 08:00 in Fig.4 (c.1), according to the bias-corrected observations the dust storm seems to have already reached central China which

was probably not the case. In comparison, the LSTM based bias-corrected dust observations (see Fig.4 (e.1)~(e.3)), which is calculated by subtracting the LSTM *non-dust* part (see Fig.4(d.1)~(d.3)) from the raw $PM_{10}$ measurements, are close to our expectations. Only for sites that are very close to the source regions high dust concentrations are derived at 08:00, while for the other sites hardly any dust is derived. At 19:00, thus 11 hours later, at half of the stations in the north of the domain high dust concentrations are derived. In the southeast of the domain, the derived

dust concentrations remain almost zero since the dust plume did not arrive there yet. At 22:00, the plume is moved further south, and the dust load closer to the source region started to decrease.

### 3.3.2 Time series

To further evaluate the two bias correction methods, Fig.5 shows the time series at the following selected cities: Hohhot, Changchun, Beijing, Baoding, Xingtai and Yulin. The location of these cities/sites can be found in Fig.1.

These cities were selected because they all experienced a severe pollution and illustrated the general performance of the LOTOS-EUROS/*non-dust* and LSTM methods. In addition, each of these cities have at least 4 monitoring sites which assured a high accuracy.

The LOTOS-EUROS grid cells with the selected sites all include other observation sites as well, and to illustrate the spread in the observations the maximum and minimum observed values in the grid cell are added to the time

series too. Similarly, the LSTM *non-dust* $PM_{10}$ simulation is given together with the spread within the grid cell.

Before the dust storm arrived at these cites, the LSTM model reproduces the variations in $PM_{10}$ rather well. Some errors are present, for example as can be seen on April 14 from 12:00 to 23:00 in Yulin. After the arrival of the dust storm, the $PM_{10}$ observations strongly increase, while the LSTM *non-dust* fraction remains at a low level since it is

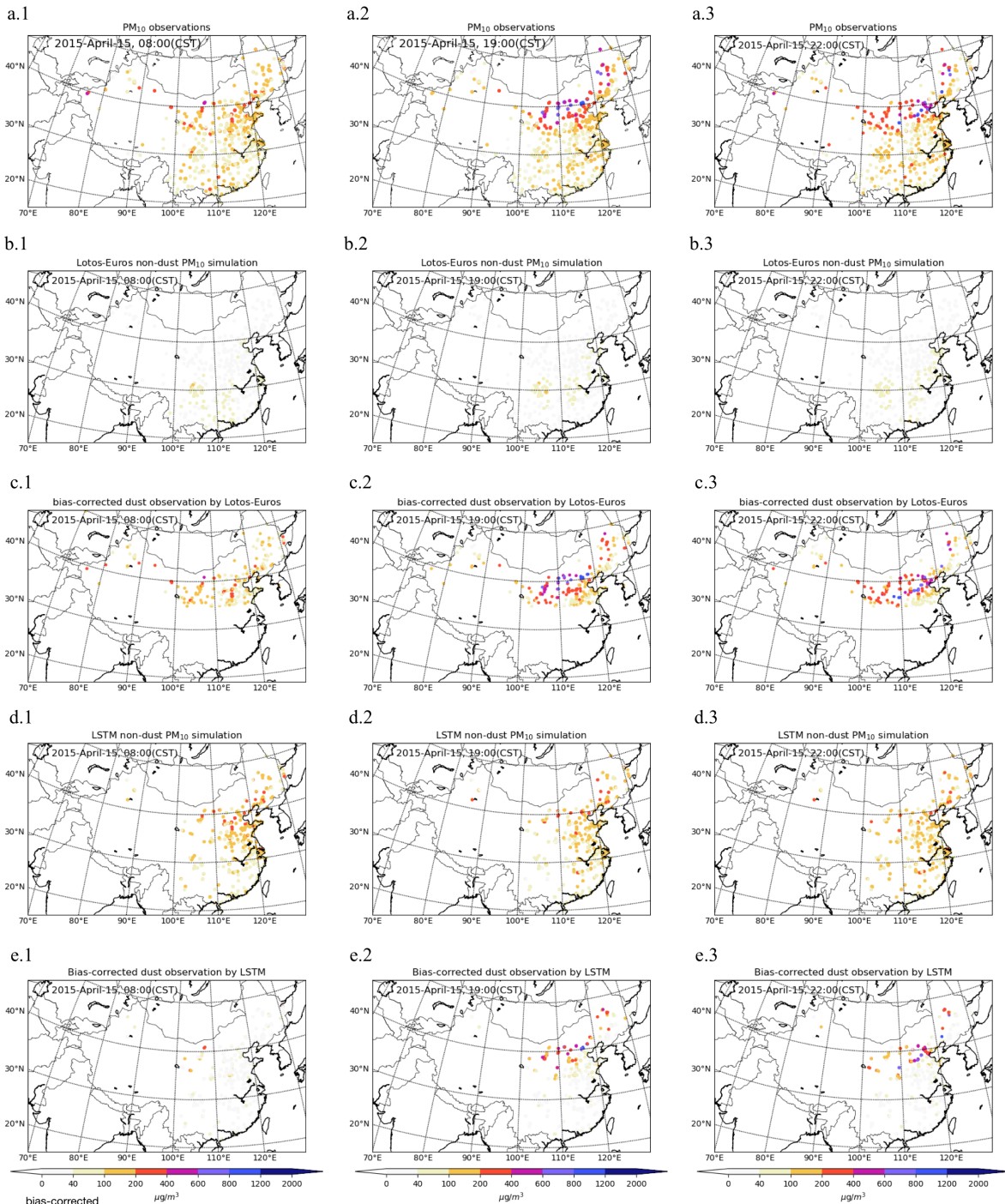

**Figure 4.** Original PM$_{10}$ measurements (a.1~a.3), LOTOS-EUROS/*non-dust* simulated PM$_{10}$ (b.1~b.3) and the corresponding bias-corrected dust observations (c.1~c.3), LSTM predicted *non-dust* PM$_{10}$ (d.1~d.3) and the derived dust observations (e.1~e.3) at three time snapshots: April 15, 08:00 (a.1~e.1), 19:00 (a.2~e.2) and 22:00 (a.3~e.3)

independent of the dust storm. The real dust measurement is then calculated by subtracting the *non-dust* part from

the raw PM$_{10}$ observations.

The LOTOS-EUROS/*non-dust* simulations underestimate the *non-dust* PM$_{10}$ at all the six locations. Thus, the derived bias-corrected dust observations overestimate the actual dust load, and this will affect the dust assimilation results.

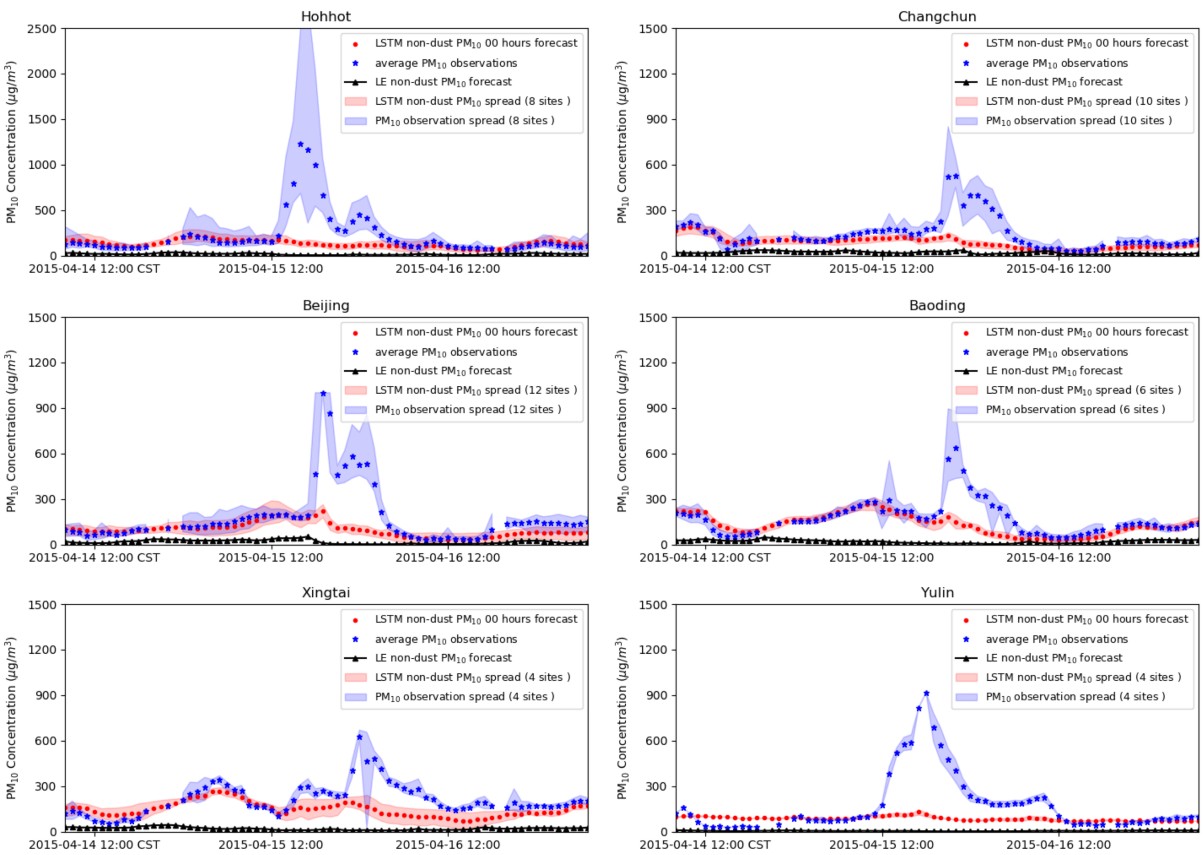

**Figure 5.** Time series of PM$_{10}$ measurements, LOTOS-EUROS/*non-dust* and LSTM predicted PM$_{10}$ levels at six cities: Hohhot, Changchun, Beijing, Baoding, Xingtai and Yulin. LE: LOTOS-EUROS; LSTM: long short-term memory.

## 4   Data assimilation experiments

Three different sets of observations are now available for assimilation in the dust model: the original PM$_{10}$ observations, the PM$_{10}$ observations with LOTOS-EUROS bias correction, and the PM$_{10}$ observations with machine learning bias correction. The results have been compared in terms of the posterior dust emission fields and surface

5   dust concentrations.

A practical use of assimilated concentrations is to use them as a start point for a forecast. This could be used to provide early information about the arrival of the dust plume and the expected dust level. The dust forecast after the end of the assimilation window at April 15 19:00 uses the newly estimated emissions. Apart from the dust concentrations, the forecast will also be evaluated in terms of skill scores for the total $PM_{10}$ concentrations in Section 4.3.

## 4.1 Observation error configuration

A key element of the data assimilation system is the observation error covariance matrix $\mathbf{R}$. This covariance quantifies the possible difference between simulations and observations. The observations with a smaller error have a higher weight in the assimilation process.

In related works, the dust observation errors were usually empirically quantified. Lin et al. (2008) assumed that the observation error is proportional to the measurement with a constant factor of 10%. Jin et al. (2018) used a similar error setting but also assigned a larger error to low valued measurements since the model might easily results in relative large errors when simulating minor dust loads.

Theoretically, the observation uncertainties are due to the representation errors as well as the measurement errors, while the former one is widely considered as the largest source. Limited by the computation resources, our dust model uses a spatial resolution of 25 km, while the in-situ measurements cover the much less of atmosphere surrounding them (Schutgens et al., 2016). This of course limits our capability of resolving the fine-scale fields that are reflected in observation spaces. Therefore, the spatial representation error is assumed to be the dominant error source and taken into the account in approximating the observation uncertainties. In addition, the error due to the different bias correction terms is indeed another source. It is not yet considered in this study but will be exploited for a more accurate assimilation operation in our future work.

The spatial representation error quantification itself is a complex task. It could be calculated through comparing the model simulations at different scales of resolutions. In this study, the availability of multiple measurement sites in a single model grid cell provides an alternative way to quantify the representation error. When multiple observations are present, the statistical error in the observed values reflects the spatial representation uncertainty. An example is the grid cell covering the city of Beijing, where observations from 12 different field stations are available. Note that it is the grid cell which has the most monitoring stations. The spread of the hourly measurements is shown in Fig.5(c). For each hour, the standard deviation of the measured $PM_{10}$ values is plotted against the mean in Fig.6, where the red markers represent 'regular' polluted conditions, and the blue markers the dust event. The result shows that the spread in the observations closely agrees with the average pollution level during the dust event. Based on this result, a simple linear regression is used to obtain a parametrization for the observation representation error:

$$\sigma = \max( \, a \cdot y \, + \, b \, , \ \ \sigma_{min} \, ) \qquad \qquad [\mu g/m^3] \qquad\qquad\qquad\qquad (9)$$

where $a = 0.12$ and $b = 55.7$ are the linear regression parameters based on the dust event data (blue markers). It should be noted that the observation sites in Beijing truncate observations at a maximum of 1000 $\mu g/m^3$, and therefore observations close to this number are not used since the true values might have been much higher. A minimum observation representation uncertainty of $\sigma_{min} = 100$ $\mu g/m^3$ is used for the 'dust' observations ($PM_{10}$ with bias correction) to avoid a too strong impact of low valued observations (hardly dust) on the estimation of dust

emissions. In case the simulation model estimates dust concentrations at the surface while in reality the plume is elevated, the low valued observations might lead to an unrealistic strong decrease of the dust emissions.

The representation uncertainty has already been validated to fluctuate in space (Schutgens et al., 2016). However, for most other grid cells the number of observations sites is simply one, which makes it difficult to parametrize a representation error in a similar way. Therefore, the representation error parametrized for Beijing is used for all

other locations too.

Note that the raw $PM_{10}$ and the bias-corrected dust measurements might have different uncertainties in representing the real dust storm level. This is not yet taken into account in our study, and the three types of the assimilated measurements, raw $PM_{10}$, bias-corrected dust observation either using the CTM or using the machine learning, are all configured with the same observation error in Eq.9. In addition, all the measurements are assumed to be

independent, hence, the observation error covariance $\mathbf{R}$ is diagonal.

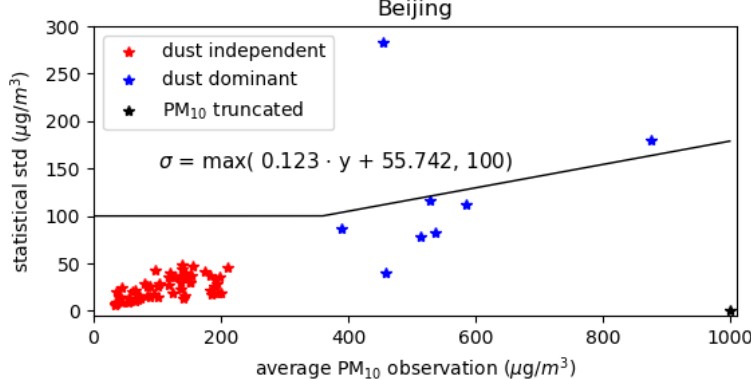

**Figure 6.** Average vs. Standard deviation of the hourly $PM_{10}$ observations range from April 14 08:00 to April 17 07:00 in the grid cell of Beijing. See Fig.5(c) for the time series.

## 4.2  Dust emission estimation

To evaluate the posterior dust emission field that is obtained by assimilation of the bias corrected 'dust' observations, an emission index $\mathcal{F}_i$ (g/m$^2$) is defined as in (Jin et al., 2018). The index represents the accumulated dust emission in a cell $i$ between April 14 08:00 and April 15 19:00. Fig.7 shows the emission index map of the *a prior* model, and

*posteriori* emissions obtained from assimilation of either the original $PM_{10}$ observations, or the LOTOS-EUROS or LSTM based bias-corrected 'dust' measurements.

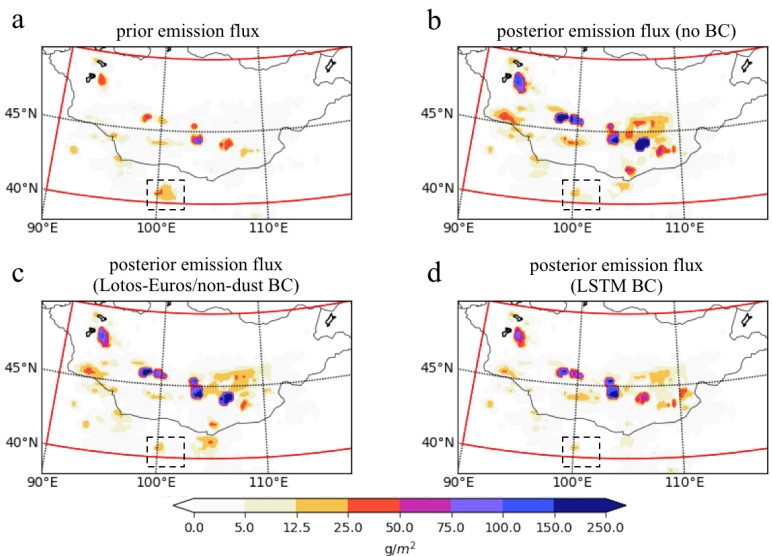

**Figure 7.** Accumulated dust emission map $\mathcal{F}$ between April 14 08:00 and April 15 19:00 of *a priori* model (a), or (b) *a posteriori* estimates using the original $PM_{10}$ observations, (c) LOTOS-EUROS or (d) LSTM based bias-corrected dust measurements. BC: bias correction

As shown in Fig.7(a), the *a prior* emission was in general rather weak, which resulted in an underestimated surface dust concentration simulation as can be seen for example in Fig.8(a.1)~(a.2). The *posteriori* emissions are almost everywhere higher than the *a prior*. An exception is the black marked region, where the *a prior* emissions are higher.
The emissions from this black-dashed region contributed to a too-early arrival of the dust peak in the model cells over Hohhot and Xingtai as shown in Fig.9(a) and (c).

Fig.7 (b) shows the emission index $\mathcal{F}$ that results from directly assimilating the original $PM_{10}$ measurements. As expected the estimated emissions are higher than those obtained by assimilating the bias-corrected observations, since all airborne aerosols observed are attributed to be dust. In comparison, the assimilation with LSTM baseline removed data results in a modest emission level as shown in Fig.7(d). The emissions estimated with LOTOS-EUROS based bias-corrected observations are in between, since the resulting 'dust' observations also overestimate the actual dust loads compared to the LSTM based bias-corrected dust measurements.

**4.3   Dust simulation and forecast skill**

Fig.8 (a)~(d) show the dust simulations at the surface layer at the end of the assimilation window (April 15 19:00, left column) and the forecast 3 hours later (22:00, right column) using the newly estimated emission field. Note that

the average dust concentration over the affected downwind regions reached at a peak around 22:00. Compared to background simulations in Jin et al. (2018), the *a prior* model simulations have been improved by disabling the topography-based preference factor as mentioned in Section 2.1; however, a large difference from the bias-corrected $PM_{10}$ observations in Fig. 4(e) is still present.

The *posteriori* concentrations in Fig.8(b.1)~(b.2) are the result of assimilating the original measurements $PM_{10}$ observations shown in Fig.4(a.1)~(a.2). As expected, these lead to the highest simulated dust concentrations since all the aerosols observed are assumed to represent dust. Especially in the center of the plume, the dust concentration can be as large as 2000 $\mu g/m^3$. Fig.8(c.1)~(c.2) show the results when using the LOTOS-EUROS/*non-dust* bias-corrected $PM_{10}$ observations as 'dust', and although concentrations are lower, they are still likely to overestimate the real dust levels. The *posteriori* results using the LSTM bias-corrected measurements provide the lowest dust concentrations as shown in Fig.8(d). Only in the grid cells that are close to the source region, the surface dust concentration reach values as large as 2000 $\mu g/m^3$, while in the downwind areas the maximum dust concentrations are usually below 1200 $\mu g/m^3$.

To illustrate the improvements of assimilating bias-corrected measurements, Fig. 9 shows the observed and simulated $PM_{10}$ concentrations in the aforementioned grid cells covering Hohhot, Beijing, and Xingtai. These locations are neither the best nor the worst examples, but illustrate typical results and challenges to be solved in future. For a fair comparison with the $PM_{10}$ observations, the non-dust aerosol concentrations obtained from either LOTOS-EUROS/*non-dust* or LSTM were added to the dust simulations from the inversion system.

Site Hohhot is close to the main dust source region. The *a prior* model simulated the arrival of the dust plume 8 hours before it was actually visible in the $PM_{10}$ observations. The assimilation of the observations is able to produce simulations in which the dust plume arrives at the correct time. The assimilation with LSTM bias-corrected data has the best performance, with the peak of the simulated concentrations (dust plus bias) most close to the observed $PM_{10}$. During the forecast period ($t$>April 15, 19:00), all three assimilation based forecasts show a decline in concentrations, which slightly overestimate the observations. This can be explained from the fact that the dust storm is a strong flow-dependent phenomenon in which concentrations at a certain location are strongly correlated to earlier concentrations at upwind locations. For Hohhot, only a limited number of observation sites is located upwind, and therefore hardly any data is available to constrain the concentrations at this location. To improve the forecast at Hohhot it will be necessary to have additional observation data, for example from sites actually within the source region, or from satellites observing the aerosol load over the source region (Jin et al., 2019).

For the grid cell Beijing, which is located further downwind from the dust source region, the arrival of the dust peak is correctly simulated. However, the amplitude of the concentration peak is underestimated compared to the average $PM_{10}$ observations. As can be seen in Fig.8, the dust plume forms a rather small band over central and northeast China. In each of the three assimilations, the dust concentrations in the band are rather low around Beijing. This suggests that the simulation model simply is not able to increase the dust concentrations here, for

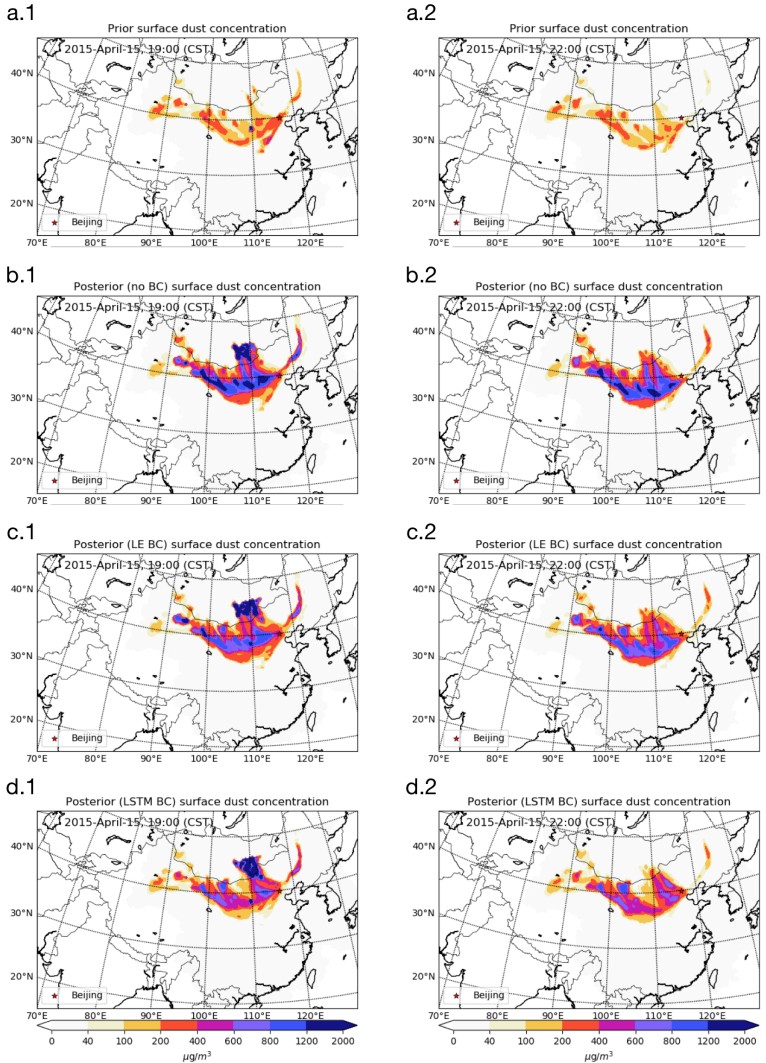

**Figure 8.** Surface dust concentration of *a prior* (a.1~a.2), *posterior* using no bias-corrected (no BC) data (b.1~b.2), *posterior* using LOTOS-EUROS/*non-dust* bias-corrected (LE BC) data (c.1~c.2), *posterior* using no bias-corrected (LSTM BC) data (d.1~d.2) at April 15, 19:00 (a.1~d.1) and 22:00 (a.2~d.2)

example because of uncertainties in the meteorological data, a removal of dust that is too efficient, or because some local sources of dust are absent (equally, non-dust $PM_{10}$ levels are underestimated).

The grid cell Xingtai is located more to the south, and the model is able to simulate high dust concentrations here. The *a prior* model simulates the arrival of a first dust peak already at 13:00, which is however not visible in the $PM_{10}$ data. The assimilation postpones the arrival of the main dust, which according to the measurements takes place around 22:00 and is already in the forecast period. The forecast simulations all overestimate the amplitude of the peak, especially when using the original $PM_{10}$ data as proxy for dust. The assimilation with the LSTM based
baseline removal shows the best agreement with the observations.

## 4.4   Evaluation of forecast skill

To evaluate the forecast skill of the assimilation(s), the root mean square error (RMSE) of the reference and three posterior full aerosol simulations (dust forecasts plus *non-dust* predictions) with respect to the observed $PM_{10}$ over the whole observation sites has been computed for each hour. A time series of this RMSE is shown in Fig.10; after
the assimilation window (marked period), the results are based on the forecast simulations. The *a prior* RMSE values at the end of the assimilation window and during the forecast are about 200-250 $\mu g/m^3$. Direct assimilation of the original $PM_{10}$ measurement actually increases these values to above 300 $\mu g/m^3$ during the forecast, since dust concentrations become strongly overestimated. Assimilation of the LOTOS-EUROS/*non-dust* baseline removed observations nonetheless reduces the RMSE, in particular within the assimilation window. Strongest decrease in
RMSE is obtained using the LSTM based baseline removal, with values of 120-200 $\mu g/m^3$ during the forecast.

## 5   Summary and conclusion

In this study, a dust storm data assimilation experiment has been performed for an event over East Asia in the spring of 2015. $PM_{10}$ observation data from the China Ministry of Environmental Protection observing network were assimilated into a dust simulation model to estimate the dust emissions. The $PM_{10}$ measurements themselves
are considered as unbiased. They clearly show the arrival of a dust plume throughout the region due to the high spatiotemporal resolution. However, the data cannot be compared directly to dust simulations since they actually represent a sum of the dust particles and other *non-dust* aerosols. Direct assimilation of these measurements would introduce a bias in the assimilation system, since it cannot distinguish between model and observation errors.

Two methods have been implemented to remove the *non-dust* part the $PM_{10}$ observations during the dust event in order to use them as 'dust' proxy in a dust assimilation system. The first method uses a conventional regional chemical
5   transport model, LOTOS-EUROS/*non-dust*, which simulates the emission, transport, chemistry, and deposition of aerosols mainly related to anthropogenic activities. The second method uses a machine learning model that statistically describes the relations between regular $PM_{10}$ concentrations (outside dust events), and available air quality and meteorological data.

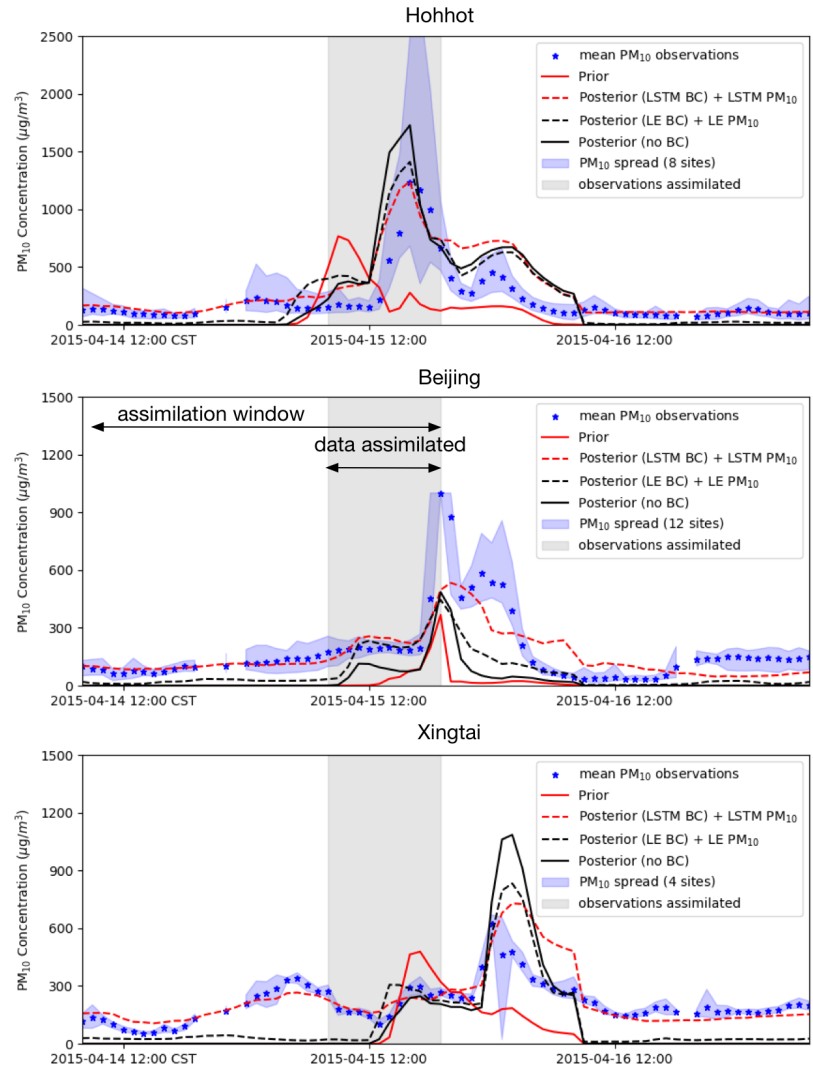

**Figure 9.** Time series of posterior dust concentration and $PM_{10}$ observations in three cities: Hohhot, Beijing, Xingtai (observations in the gray shadow are assimilated)

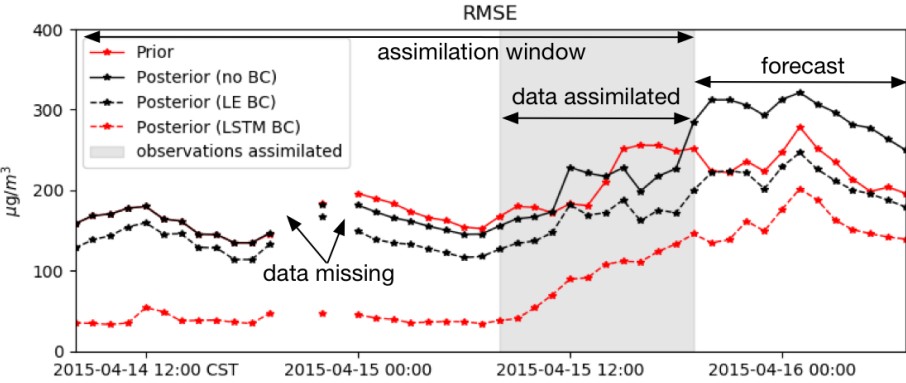

**Figure 10.** Time series of root mean square error compared to the ground PM$_{10}$. The assimilation window is set from April 14 08:00 to April 15 19:00, and PM$_{10}$ observation in the gray shadow are assimilated.

The two methods to estimate the *non-dust* part of the PM$_{10}$ load have been validated. The simulations by the LOTOS-EUROS/*non-dust* model in general underestimate the PM$_{10}$ concentrations. The root mean square error stays at a relative high level of 89.4 $\mu$g/m$^3$. It is mainly caused by missing emissions and aerosol components such as secondary organic matter. In comparison, the data-driven machine learning model agrees more closely with the real measurements, the RMSE declines to 58.6 $\mu$g/m$^3$.

A variational data assimilation system has been used to estimate the dust emissions that lead to a severe dust storm in April 2015. The system either assimilated the original PM$_{10}$ observations, or the bias-corrected 'dust' observations based on either LOTOS-EUROS/*non-dust* or LSTM model. The posterior simulations using the original observations resulted in a strong overestimation of the dust concentrations, since all PM$_{10}$ are simply attributed to dust. Using the LOTOS-EUROS/*non-dust* bias-corrected observations, a clear improvement on the dust simulation has been obtained, but overestimation of dust concentrations is still present. The best results are obtained when using a LSTM model to remove the *non-dust* part of the PM$_{10}$ observations, with *posterior* concentrations in good agreement with the measurements.

The dust emissions estimated using the assimilation can be used to drive a dust forecast. When the original PM$_{10}$ observations were used in the assimilation, the forecast skill of the system actually decreased due to the strong overestimation of dust concentrations, the RMSE rose from averagely 230 (prior forecast) to 300 $\mu$g/m$^3$. Better forecasts are obtained when using the model-based and especially the machine learning based bias-corrected observations. The RMSE of the former one was reduced to 200 $\mu$g/m$^3$ while the RMSE of the latter one further declined to 150 $\mu$g/m$^3$.

### Future work

Both our CTM and machine learning based bias correction methods have room for improvements. It might be useful to improve the CTM simulations by assimilating $PM_{10}$ observations during the hours where no dust storms are present, and use these improved simulations to remove the *non-dust* part of the observations during an event. These additional assimilations would then involve repeated forward ensemble bias-model runs which could be computationally expensive. The machine learning model in our *non-dust* $PM_{10}$ simulation can also be further optimized, such as using a different configuration or deeper neural network, including extra input features like *non-dust* $PM_{10}$ simulation from CTMs (Lin et al., 2019) and other related records.

We will exploited the variabilities of the representation errors comparing the model simulations at different spatial resolutions. The error from the bias correction term will also be taken into account while calculating the observation error.

### Data availability

The datasets including measurements and model simulations can be accessed from websites listed in the references or by contacting the corresponding author

### Author contribution

JJ and HXL conceived the study and designed the experiments. JJ and YX performed the machine learning based non-dust simulation. AS performed the CTM based non-dust simulation. JJ and AS performed the assimilation tests and carried out the data analysis. AS, HXL, AH provided useful comments on the paper. JJ prepared the manuscript with contributions from all co-authors.

### Acknowledgments

The real-time $PM_{10}$ data are from the network established by the China Ministry of Environmental Protection and accessible to the public at http://106.37.208.233:20035/. One can also access the historical profile by visiting http://www.aqistudy.cn/.

### Competing interests

The authors declare that they have no conflict of interest.

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
