# Peer review of "Machine learning for observation bias correction with application to dust storm data assimilation"

_Atmospheric Chemistry and Physics, 2019_

## Referee Comment (RC1) · Anonymous Referee #1 · 12 Jun 2019

The main purpose of this paper is to explore the impact of observational biases on dust assimilation. These biases exist because surface PM10 contains both dust and non-dust contributions. However, the authors discuss two methods to separate out dust PM10 and then use it in a data assimilation scheme. Novel is their use of a machine learning technique (a neural net called LSTM) to do this. The authors show that, if not corrected, such biases can seriously (and negatively) affect dust forecasts but that use of the LSTM allows for significantly better forecasts. The paper is fairly technical and may be better suited to GMD but is appropriate for ACP too. The paper is well-written, properly organised with good graphics. Its conclusions are warranted by the data presented.

My main comment would be to add more detail to the section on the LSTM bias cor-

rection.

p 2, l 21: please consider adding "Inverting East Asian Dust Emissions of a Severe Winter Dust Event Using the Ensemble Kalman Smoother and Himawari-8 AODs" by Tie Dai, currently in review

p 3, l 25: it is a pity the authors do not compare the correction scheme in Eq (1) also. Maybe it is too much work to do the full assimilation work but at least a comparison of non-dust PM10 estimates should be possible?

p 7, l 5: What sort of uncertainty in emissions results from this: 1%, 10%, 100%? What is the spatial structure of delta f? Does i vary freely from grid-box to grid-box or is some correlation imposed?

p 10, l 10: why are PM25 measurements at nearby (!) sites used? The authors already used PM25 from the AQ network. What sort of error do they think is introduced by using measurements taken at different locations?

p 10, l 20: I would not say the "actually are" non-dust PM10 but rather that they may be assumed to be very close to non-dust PM10

p 10, l 22: "measurements at the nearby sites are considered as the essential input". Maybe the authors can clarify?

p 10, Sect 3.2: since this is arguably the new contribution by this paper, more details seems required. E.g. 1) how large where training and testing datsets; 2) how where they separated?; 3) how where LSTM's hyper-parameters set? 4) Please provide error statistics for training & testing. I am familiar with LSTM use in time-series but in this paper the authors also claim it is useful for spatial variations (p 10, l 1). Was the LSTM adapted to allow for this? It seems random forests might equally be useful. Did the authors consider this? Random forest is easier to train and would also provide feedback on what are the most relevant features for non-dust PM10 prediction.

p 11, l 22: why use a t=1 forecast to estimate non-dust PM10. Can't you train the LSTM

to estimate non-dust PM10 at t=0?

p 16, l 1: Would R vary due to bias correction scheme? Afterall that scheme will affect remaining errors in dust PM10. In particular, I wonder whether the LSTM scheme (which trains on all data throughout a domain) would not lead to off-diagonal elements to be non-zero (i.e. correlations amongst errors at nearby stations)? Please discuss.

p 16, l 5: Can the authors discuss the largest sources for the errors of 10%? Measurement error? non-dust correction? Spatial representation? I see the authors suggest (!) representation is the most important but they give no reason for this. non-dust correction errors are of course available from the LSTM training.

p 16, l 15: "and thus the representation uncertainty" I do not agree with this. The representation error is the difference between a point measurement and a grid average. Here the authors consider variation among point measurements, which is likely to over-estimate representation error. Also, representation error will can show significant variation depending on location of point measurement in grid box, see e.g. Schutgens et al. ACP 2016.

p 16, l 16: Can the authors surmise why relative standard deviation (std/mean) is almost constant for non-dust and dust event? Of course, PM10 levels increase but wouldn't one expect a smoother field in case of a dust plume (long range transport) vs local pollution?

p 16, l 27: Would the authors not expect the representation error in an Urban area to be signiifcantly larger than in the countryside? Also, shouldn't representation error really be calculated for the dust PM10, instead of the total PM10?
* * *

---

## Referee Comment (RC2) · Anonymous Referee #2 · 13 Jun 2019

This article looks at the effect of applying a bias correction term to PM10 observations in order to assimilate them in to a dust model as 'dust' observations. The bias correction is needed since PM10 observations account not just for dust, but also other types of aerosol. The paper compares two different bias correction methods to just assimilating the PM10 observations directly. The first bias correction method is to use a CTM to calculate the non-dust part of the PM10 observations and the second more novel method is a machine learning approach.

The paper is in general well written with clear figures, although there are a few grammatical English errors (those that I could easily comment on are listed under typos below). The structure is straight forward and the logic easy to follow. It is also fairly comprehensive and so provides an in-depth examination of the two different bias cor-

rection methods. However, it does not feel that this is too much information but rather provides context as to why the assimilation results are seen.

I have two main comments to make on the article. The first is that it wasn't clear to me, from reading through the article, how such a bias correction would be applied in a real life scenario. There is some comment on this in the summary and conclusions but it would be interesting and very relevant to understand how either bias correction scheme might be applied in practice. It is not obvious to me how this would be possible. My second comment is that I sometimes struggled to understand the detail of what had actually been done in each of the two schemes and this is addressed in the more specific comments below:

Comments:

Pg. 1, line 20. I don't believe that the conclusion can be drawn in general that the best results are obtained when using a machine learning model. This is true in this article using this CTM, but the results that show an under-estimation of the non-dust PM10 from the CTM would, to me, provide evidence that the CTM is a slightly flawed model for this aspect and hence why the machine learning approach performs better. It would be enough to change this sentence to 'The best results are obtained when using the machine learning model...'

Pg. 7, line 7. What do you mean to release the efforts in updating the tangent linear model? To me this would imply the effort to keep the tangent linear model up to date with the full non-linear model, but I'm assuming that what you really mean is to reduce the time spent in calculating the tangent linear model?

Pg. 9, Section 3.1. I assume this chemical transport model was chosen because it produces a full operational forecast over the modeling domain? Does this therefore make the application of the bias correction possible for a real life scenario? If true, then this would provide a good explanation for why this model is used to evaluate between the two different schemes when it is clearly unable to estimate the non-dust PM10

observations.

Pg. 10, line 10. Are the input observations of PM25, SO2, NO2, O3 and CO all coming from the same station as the PM10 observations used to compare the output? Is this why the nearby sites of PM25 are included as a separate line. Does the machine learning model therefore come up with a different value of PM10 for each station separately or is all the data included together so that given the input observations from any station, a generic PM10 output would be generated?

Pg. 10, eqn 8. What does the 'm' stand for in this equation. Is it observation across the full domain and time period or just observations at one station over the time period?

Pg. 11, line 22-line 28. The description in this paragraph needs further clarification. I believe that you are subtracting the 1-hr forecast made from April 15, 19:00 (so valid at 20:00?) from all the PM10 observations between 8:00 and 19:00 for assimilation. Or is it the one hour forecast valid at each specific time of the observation? In which case does each observation from each station require different 18hour input? Why is the 1-hr forecast chosen rather than the actual value coming from the machine-learning method? How is a forecast made from a machine learning method?

Similarly, is the 12-hr forecast from April 15, 19:00 (so valid at April 16, 07:00) added to all the forecast dust PM10 values to compare to the observations?

Pg. 9-11, Section 3.2. How would such a machine learning algorithm be used in practice? Would it need the constant evaluation of running 18hrs of data? Or if a dust storm was forecast, would the calculations be switched on. At what point would the PM10 observations be assimilated. Or is this envisaged mode for a reanalysis style product of PM10 observations?

Pg. 12, line 9-10. You state that the according to the LOTOS-EUROS bias corrected observations, the dust storm seems to have already reached central China which was in reality not the case? How do you know this? Nothing from this figure provides

evidence of what the reality was? It would be worth referencing how you know this to be true.

Pg. 13, line 17-19. This to me provides evidence that the LOTOS-EUROS model is not doing a good job of predicting the non-dust PM10 and so is never going to match the performance of the machine learning algorithm. Is there a reason for choosing this model or not exploring CTMs that may provide a better match?

Pg. 18, line 20. Again, you state that the LOTOS-EUROS observations over-estimate the observations and yet I can't see anything in Figure 8 that shows the observations, so that we know this.

Pg. 18, line 31-33. It is interesting to me that although the peak is a better match with the machine learning algorithm, the forecast is actually slightly better with the LOTOS-EUROS model. Do you have any feeling for why this may be?

Typos:

Pg. 1, line 13. I think this should read 'The latter is trained by learning using two years of historical samples'.

Pg. 2, line 9. Should be 'progress has also been made'

Pg. 2, line 19. Should be 'A wide variety of data assimilation techniques have been used...'

Pg. 2, line 28. Should be 'In the presence of biases...'

Pg. 10, line 25. Should be 'Earlier studies showed that the...'

---

## Referee Comment (RC3) · Anonymous Referee #3 · 14 Jun 2019

The aim of this work is to develop a methodology able to use PM10 as a tracer of dust with the last end of assimilating PM10 to improve the forecast of dust storms. The deterrent of direct assimilation of PM10 is that it does not only encompass dust but also other aerosol flavors that are more significant where anthropogenic sources of aerosols are present. To isolate dust and non-dust aerosols the idea is to consider non-dust as bias in the PM10 records. Then the problem is reduced to apply bias correction methods to PM10. Here the authors propose to use machine learning and chemical transport model for bias correction. Results show a more accurate forecast of dust storm if the PM10 assimilated is isolated form non-dust aerosols using the machine learning method.

The paper is well written, structured and results support the conclusion archived. The

topic is innovative since it is one of the first works to apply machine learning to bias correction to isolate non-dust aerosols from PM10. In addition, the methodology developed here goes beyond to improve the forecast of dust storm and it can benefit the assimilation of other magnitudes. This study match with the ACP topics and its quality is close to its standards.

However, there are some general comments that need to be addressed or clarified along with the manuscript.

1) How this approach guarantees the "bias" in PM10 that you are correcting is non-dust and it is not a "real" bias present in your PM10 records?

2) How do you know a better performance of LSTM for bias correction (or dust isolation from PM10) and also better dust forecast if you are not comparing against dust records? I am aware of the lack of availability of them but at least it will be necessary to actually know that your PM10 without bias is dust by comparing with some station or a satellite image.

3) Also, the main caveat of the machine learning methods is the availability of a long series of records and the computational time. Do you think that this method will be able to correct the PM10 bias obtained a few hours (e.g. 12 h) before a dust storm and be ready to assimilate and perform the forecast a few hours later? The real applicability of LSTM is not clear. This point needs a little more discussion.

4) Finally, the conclusion about what method is better for bias correction looks like that is the same method that better "simulate" the non-dust aerosols. Then, the worst bias correction with LOTOS-EUROS model is just the worst performance in reproducing non-dust aerosols. This simplification is not straightforward needs to be clarified.

Minor comments

P4-L29. to large -> too large

P5-L2. Can easily applied -> can easily be applied

P7-L3. Following Jin et al., 2018

P9-L15. Anthropogenic emissions are from MEIC but it is not clear where natural emissions are obtained. Are these natural emissions; sea salt, biogenic emissions, and wildfires, but not dust?

P10-L11 How LSTM is used needs little more details. It is not clear why these elements of the input vector have been chosen. Do these observations correspond to blue dots in Fig. 1? Why do you use nearby sites for PM2.5 but not for the other species? Is the training period only past 18 h or from January 2013 to March 2015 as it is said in P11-L4? Are these data available hourly? P11-L23 What criteria do you use to consider that a series has high data missing?

P11-L4 I would like to see (or at least discuss) the sensitivity of the results to the training period. For instance, in the manuscript it is argued that from 01-2013 to 03-2015 the frequency of dust storm was low then it could be considered that all PM10 were non-dust. However, yb will be significantly larger during the days with a dust storm and this could affect the regression model. Have you checked if the regression model changes if you remove those days with dust storm?

P12-L12 In non-dust PM10 evaluation, how do you know that they are dust in Fig. 3? Is it only based on larger numbers?

P11-L1 How do you know LE/non-dust underestimate non-dust PM10? Is it based in Fig. 3 and lower values of PM10?

P11-L5 To show the better agreement of LSTM than LE/no-dust, could you give some number such as correlations?

P13-L25. Is a high level of pollution the reason why these sites are chosen to test the performance of LE/non-dust?

P20-L15 In addition to the orography, a reason why in Beijing the PM10 is underestimated could be related to higher PM10 non-dust values? Did you find any relationship

between more polluted cities and worst performance of the bias correction?

In the section of the evaluation of the forecast skill, I miss comparing with "real" dust records instead of PM10.

Figures, in general, are clear and the captions properly describe them.

Fig.1 It is difficult to distinguish the star symbols which denote the cities location. Also, Is N=1352 number of stations used in LSTM?

Fig. 3 Use properly notation for float numbers (e.g. 2 10-6 instead of 0.00002).

Fig. 6 Unify the legends and try it does not overlap with the time series.

---

## Author Comment (AC1) · 12 Jul 2019

**Response to Referee #1:** We would like to thank the referee for the careful review and thoughtful suggestion, especially the suggestion of replacing the t=1 bias forecast with that of t=0, which helps us further improve the quality of the manuscript.

Our response follows (*the reviewer's comments are in italics*)

*General comments:*
*The main purpose of this paper is to explore the impact of observational biases on dust assimilation. These biases exist because surface PM10 contains both dust and non-dust contributions. However, the authors discuss two methods to separate out dust PM10 and then use it in a data assimilation scheme. Novel is their use of a machine learning technique (a neural net called LSTM) to do this. The authors show that, if not corrected, such biases can seriously (and negatively) affect dust forecasts but that use of the LSTM allows for significantly better forecasts. The paper is fairly technical and may be better suited to GMD but is appropriate for ACP too. The paper is well-written, properly organised with good graphics. Its conclusions are warranted by the data presented.*

*My main comment would be to add more detail to the section on the LSTM bias correction.*
**Reply**: We agree with the referee that more details about LSTM are necessary, more information is now added to describe the LSTM bias correction in **Section. 3.2 Machine learning for non-dust PM10 simulation** on page 10-12.

**Question 1:** *p 2, l 21: please consider adding "Inverting East Asian Dust Emissions of a Severe Winter Dust Event Using the Ensemble Kalman Smoother and Himawari-8 AODs" by Tie Dai, currently in review.*
**Reply**: We are very interested in this relevant paper. We would appreciate if the referee could provide us the link of source paper/citation (at the present we are unable to find the article).

**Question 2:** *p 3, l 25: it is a pity the authors do not compare the correction scheme in Eq (1) also. Maybe it is too much work to do the full assimilation work but at least a comparison of non-dust PM10 estimates should be possible?*
**Reply**: Thanks for pointing out this issue. We already used this method (Eq.1) in our previous paper (Jin et al., 2018). This method worked fine when the relation between PM2.5 and PM10 in a given observation site exhibited a simple linear correlation. However, in many sites the application of Eq.1 failed when the linear correlation (**R**) between PM2.5 and PM10 is weak. Actually, in that study we only performed the Eq.1 for the site when **R** ($PM_{2.5}$, $PM_{10}$) > 0.8 (accuracy is promised). In that case, observations in 45% sites are abandoned. Unlike the machine learning method which can be used almost everywhere.
To point out the weakness of Eq.1, we added the remarks on page 3, line 30-32 "***For instance, in the dust event studied in Jin et al. (2018), the application of Eq.1 in many sites failed since R is weak. To have a quality-assured bias correction, Eq.1 is performed only when the Pearson correlation coefficient R > 0.8. Consequently, measurements in around 45% sites are rejected in that case.***"

**Question 3:** *p 7, l 5: What sort of uncertainty in emissions results from this: 1%, 10%, 100%? What is the spatial structure of delta f? Does i vary freely from grid-box to grid-box or is some correlation imposed?*

*Reply:* The emission uncertainty is assumed to result from the uncertainty of friction velocity threshold in the windblown parametrization. We add more explanation on page 7, line 5-10 "***Following Jin et al. 2018, the errors in dust emission field were assumed to be only caused by the uncertainty in the friction velocity threshold in the dust windblown parametrization, and similar assumptions on the uncertainty are used to build an emission error covariance B. The friction velocity threshold is perturbed with a spatially varying multiplicative factor β with a mean of 1 and a standard deviation of 0.1. In addition, an exponential profile of distance-based spatial correlation is posed on βs (Jin et al., 2018)***.

*Question 4:* p 10, l 10: why are PM25 measurements at nearby sites used? The authors already used PM25 from the AQ network. What sort of error do they think is introduced by using measurements taken at different locations?

*Reply:* Yes, the reason why the air quality observations (not only PM2.5) at nearby sites are used are indeed not clearly clarified. We now add some contexts to explain the necessities by saying "***For the non-dust PM10 machine learning forecasts in a given site, observations from its nearby sites are also vital and are used in two ways. First, missing data records are unavoidable in an air quality monitoring network, while the LSTM model training requires an uninterrupted time series of features. In this study, data interpolations of air quality measurements (PM10, PM2.5, SO2, NO2, O3 and CO) are performed using both a linear interpolation and a k-Nearest-Neighbor algorithm (Zhang, 2012) if a site has no more than 30% of missing data. Otherwise, all the measurements in the given sites are abandoned. Generally, more information available from the nearby sites will result in a more accurate interpolation. Second, learning in the presence of data errors is pervasive in machine learning, and the measurements from nearby stations are used to limit their influence. Data errors occur due to incorrect sensor readings, software bugs in the data processing pipeline, or even the inaccurate data interpolation. Statistical analysis tests have been conducted which did not only indicate a strong correlation between the non-dust PM10 and air quality measurements in the given sites, but also show that the predictor (non-dust PM10) is correlated to the observation indices (especially the PM2.5) at its nearby sites. In order to eliminating errors caused by incorrect inputs at the modeling site, the measurements at the nearby stations are considered as the essential indices. In this study, a data instance will only be selected for training the LSTM model if there is at least one nearby site within an empirical radius 0.8\degree (approx 80 km), and a maximum of 3 nearby sites will be randomly selected where observation stations are densely distributed. To save the computation costs on machine learning model training, only the PM2.5 from the nearby sites are included as one of the inputs in this study.***" on page 11, line 20-34.

*Question 5:* p 10, l 20: I would not say the "actually are" non-dust PM10 but rather that they may be assumed to be very close to non-dust PM10.

*Reply:* We agree with the referee. "***actually are***" is now changed to "***very close to***" on page 11, line 12.

*Question 6:* p 10, l 22: "measurements at the nearby sites are considered as the essential input". Maybe the authors can clarify?

*Reply:* See the **Reply** to **Question 4**.

*Question 7:* p 10, Sect 3.2: since this is arguably the new contribution by this paper, more details seems required. E.g. 1) how large where training and testing datsets; 2) how where they separated?; 3) how where LSTM's hyper-parameters set? 4) Please provide error statistics for training & testing. I am familiar with LSTM use in time-series but in this paper the authors also claim it is useful for spatial

*variations (p 10, l 1). Was the LSTM adapted to allow for this? It seems random forests might equally be useful. Did the authors consider this? Random forest is easier to train and would also provide feedback on what are the most relevant features for non-dust PM10 prediction.*

**Reply:** Regarding the training and testing dataset: It was not clearly illustrated, we now added more remarks on page 11, line 4-6 "***The training dataset covers the period from January 2013 to March 2015. In other words, the LSTM model L is trained to best fit the samples from this period. The two months April and May 2015 in which the studied dust event occurred is set as the testing period.***"

Regarding how training/testing period are separated: We did not consider this separation too much, like the 8:2 (or 7:3) ratio between the training and testing dataset. The air quality measurements that we have started from 2013. The dust storm we studied is in 2015 April. So, we set the training period from 2013 Jan to 2015 Mar, while the following two months in which the dust event occurred in set as the testing periods.

Regarding the hyper-parameters set: Hyper-parameters are listed in Table.1 LSTM hyper-parameters on page 12.

Regarding the error statistics for training and testing: Both statistic errors from the CTM and Machine learning are now presented. RMSE of CTM are given on page 12, line 21-22 by saying "***The CTM LOTOS-EUROS/non-dust in general underestimates the non-dust PM10. The forecast results in a relatively large root mean square error (RMSE) 89.4 ug/m3.***"; RMSE of the two LSTM are given on page 12, line 24-27 by saying "***The two LSTM forecasts show on average a good agreement with the observations. The RMSEs of the forecasts of the two machine learning models in the two years of training period are reduced to 55.9 and 60.7 ug/m3, and in the two months of test period (excluding the dust event April 14 to 16) they also stay at comparable low levels of 58.6 and 60.2 ug/m3.***"; These RMSEs are also shown in Fig.3 on page 13.

Regarding "***the LSTM is useful for spatial variation***": It is a typo error, actually we performed the LSTM training and testing/forecast site by site as we mentioned on page 12, line 1-3 "***The machine learning model for non-dust PM10 forecast is trained site by site, with the hyper-parameters shown in Table.1***". Also on page 10, line 16 "***in predicting spatial and temporal varying time series problem***" is corrected to "***in predicting time series problem***".

Regarding the option of random forests: Thanks for the suggestion. The random forests will be valuable when there are various input features but only parts of them are relevant to the output. We did not consider it yet, but will take it into account in our future work.

**Question 8:** *p 11, l 22: why use a t=1 forecast to estimate non-dust PM10. Can't you train the LSTM to estimate non-dust PM10 at t=0?*

**Reply:** Thanks for this suggestion. We agree with the referee that t=0 simulation could indeed replace the t=1 forecast, and are expected to give a better non-dust PM10 simulation for bias correction. New bias simulations "t=0" have been performed, as well as assimilation runs with the new bias-corrected dust observations. Both of the bias-correction as well as the posterior dust forecast using t=0 indeed showed some improvements. New results are now used to replace the previous ones, they can be mainly found in Fig.3b (non-dust simulation vs. PM10); ***Fig.4d-4e*** (spatial distribution of the non-dust level and bias corrected dust observations); ***Fig.5*** (time series in 6 cities); ***Fig.7d*** (posterior emission field by assimilating bias-corrected data using LSTM); ***Fig.8d*** (posterior of dust simulation using the bias-corrected measurement by LSTM); ***Fig.9*** (posterior dust forecast in 3 cities) and ***Fig.10*** (RMSE). From the perspective of RMSE, t=0 provides a slightly better result.

**Question 9:** *p 16, l 1: Would R vary due to bias correction scheme? Afterall that scheme will affect remaining errors in dust PM10. In particular, I wonder whether the LSTM scheme (which trains on all data throughout a domain) would not lead to off-diagonal elements to be non-zero (i.e. correlations amongst errors at nearby stations)? Please discuss.*

**Reply:** All the three types of assimilation tests used the same $R$, and all the observation assimilated are assumed to be independent (since the LSTM is performed site by site). We agree with the referee that the three types of data might have different $R$ since they have different remaining biases. However, it will then be difficult to judge/evaluate whether the adding values of new assimilation is achieved by using the bias correction or new observation error covariance.

We added the remarks to point out the possibility of using different $R$ by saying "***Note that the raw PM10 and the bias-corrected dust measurements might have different uncertainties in representing the real dust storm level. This is not yet taken into account in our study, and the three types of the assimilated measurements, raw PM10, bias-corrected dust observation either using the CTM or using the machine learning, are all configured with the same observation error in Eq.9. In addition, all the measurements are assumed to be independent, hence, the observation error covariance R is diagonal.***" on page 18, line 12-16.

**Question 10:** *p 16, l 5: Can the authors discuss the largest sources for the errors of 10%? Measurement error? non-dust correction? Spatial representation? I see the authors suggest (!) representation is the most important but they give no reason for this. non-dust correction errors are of course available from the LSTM training.*

**Reply:** We assume the spatial representation error is the largest source in this study. More remarks are given to explain this point by saying "***Theoretically, the observation uncertainties are due to the representation errors as well as the measurement errors, while the former one is widely considered as the largest source. Limited by the computation resources, our dust model uses a spatial resolution of 25 km, while the in-situ measurements cover the much less of atmosphere surrounding them (Schutgens et al. 2016). This of course limits our capability of resolving the fine-scale fields that are reflected in observation spaces. Therefore, the spatial representation error is assumed to be the dominant error source and taken into the account in approximating the observation uncertainties. In addition, the error due to the different bias correction term is another source. It is not considered yet in this study but will be exploited for have a more accurate assimilation test in our future work.***" on page 17, line 14-21.

**Question 11:** *p 16, l 15: "and thus the representation uncertainty" I do not agree with this. The representation error is the difference between a point measurement and a grid average. Here the authors consider variation among point measurements, which is likely to over-estimate representation error. Also, representation error will can show significant variation depending on location of point measurement in grid box, see e.g. Schutgens et al. ACP 2016.*

**Reply:** Yes, the representation error could be calculated by comparing the model simulation at different spatial resolutions. We have now pointed this out on page 17, line 22-25, "***The spatial representation error quantification itself is a complex task. It could be calculated through comparing the model simulations at different scales of resolutions. In this study, the availability of multiple measurement sites in a single model grid cell provides an alternative way to quantify the representation error. When multiple observations are present, the statistical error in the observed values reflects the spatial representation uncertainty.***"

What we did in Fig.6 is to present the hourly average vs. stand deviation of hourly observations in the grid cell of Beijing. The standard deviation is then used to approximate the observation representation error.

We also agree with the reviewer that the representation error might vary in space and time. But in most of the grid cells, we only have one or two monitoring sites, which make it difficult to parameterizing the representation error for other grid cells in a similar way as for Beijing. To explain this, we added some new remarks on page 18, line 8-11 by saying "***The representation uncertainty has already been validated to fluctuate both in space (Schutgens et al., 2016). However, for most other grid cells the number of observations sites is simply one, which makes it difficult to parametrize a representation error in a similar way. Therefore, the representation error parametrized for Beijing is used for all other locations too.***"

*Question:* p 16, l 16: Can the authors surmise why relative standard deviation (std/mean) is almost constant for non-dust and dust event? Of course, PM10 levels increase but wouldn't one expect a smoother field in case of a dust plume (long range transport) vs local pollution?
*Reply:* Yes, it is true that the relative **std** of dust period and non-dust period in Fig.6 looks quite constant. But it is not easy to draw a general conclusion since we only have a few samples.

*Question:* p 16, l 27: Would the authors not expect the representation error in an Urban area to be significantly larger than in the countryside? Also, shouldn't representation error really be calculated for the dust PM10, instead of the total PM10?
*Reply*: Regarding the different representation error in the urban area and countryside: In the future, we would use a model of higher resolutions that helps to quantify the dynamic spatial representation error. Regarding the different representation errors: It comes to the same question (*Question 9*) that whether we should use the same *R* for different assimilated data (PM10, bias-corrected dust observation either using CTM or machine learning).

[revised manuscript text omitted]

---

## Author Comment (AC2) · 12 Jul 2019

**Response to Referee #2:** We would like to thank the referee for the careful review and the thoughtful comments, which help us to further improve the quality of the manuscript.

Our response follows (*the reviewer's comments are in italics*)

***General comments:***
*This article looks at the effect of applying a bias correction term to PM10 observations in order to assimilate them in to a dust model as 'dust' observations. The bias correction is needed since PM10 observations account not just for dust, but also other types of aerosol. The paper compares two different bias correction methods to just assimilating the PM10 observations directly. The first bias correction method is to use a CTM to calculate the non-dust part of the PM10 observations and the second more novel method is a machine learning approach.*

*The paper is in general well written with clear figures, although there are a few grammatical English errors (those that I could easily comment on are listed under typos below). The structure is straight forward and the logic easy to follow. It is also fairly comprehensive and so provides an in-depth examination of the two different bias correction methods. However, it does not feel that this is too much information but rather provides context as to why the assimilation results are seen.*

*I have two main comments to make on the article. The first is that it wasn't clear to me, from reading through the article, how such a bias correction would be applied in a real life scenario. There is some comment on this in the summary and conclusions but it would be interesting and very relevant to understand how either bias correction scheme might be applied in practice. It is not obvious to me how this would be possible. My second comment is that I sometimes struggled to understand the detail of what had actually been done in each of the two schemes and this is addressed in the more specific comments below:*

*Comments:*
***Question 1:*** *Pg. 1, line 20. I don't believe that the conclusion can be drawn in general that the best results are obtained when using a machine learning model. This is true in this article using this CTM, but the results that show an under-estimation of the non-dust PM10 from the CTM would, to me, provide evidence that the CTM is a slightly flawed model for this aspect and hence why the machine learning approach performs better. It would be enough to change this sentence to 'The best results are obtained when using the machine learning model...'*
***Reply****:* Accepted. *"**a machine learning model**" is now changed to "**the machine learning model**" on* page 1, line 20.

***Question 2:*** *Pg. 7, line 7. What do you mean to release the efforts in updating the tangent linear model? To me this would imply the effort to keep the tangent linear model up to date with the full non-linear model, but I'm assuming that what you really mean is to reduce the time spent in calculating the tangent linear model?*
***Reply****:* Thanks for point out this misleading sentence. *"**To release the efforts in updating the full tangent linear model**" is now changed to "**To reduce the computational costs in calculating the tangent linear model**" on page 7, line 12.*

***Question 3:*** *Pg. 9, Section 3.1. I assume this chemical transport model was chosen because it produces a full operational forecast over the modeling domain? Does this therefore make the application of the bias correction possible for a real life scenario? If true, then this would provide a good explanation for why this model is used to evaluate between the two different schemes when it is clearly unable to estimate the non-dust PM10 observations.*

***Reply:*** Yes, this CTM we used to model non-dust aerosol levels is the same as the operational one, but the desert dust emissions are disabled.

***Question 4:*** *Pg. 10, line 10. Are the input observations of PM25, SO2, NO2, O3 and CO all coming from the same station as the PM10 observations used to compare the output? Is this why the nearby sites of PM25 are included as a separate line. Does the machine learning model therefore come up with a different value of PM10 for each station separately or is all the data included together so that given the input observations from any station, a generic PM10 output would be generated?*

***Reply****:* We actually estimate the non-dust aerosol simulation site by site. The inputs are the PM25, SO2, NO2, O3 and CO coming from the same site, plus the PM2.5 from the nearby sites. The output is the non-dust PM10 in the given site. The machine learning model is trained to output the value that can best fit the PM10 records in the training period.

To make it clear, we added the new remarks that on page 12, line 3, "***The machine learning model for non-dust PM10 forecast is trained site by site***". In addition, we added a new paragraph on page 11, line 20-34 to explain why the measurements at the nearby site are important.

***Question 5:*** *Pg. 10, eqn 8. What does the 'm' stand for in this equation. Is it observation across the full domain and time period or just observations at one station over the time period?*

***Reply:*** Following the **Question 4**, we actually train and use the machine learning model for each site separately, the "***m***" here only represents the observations at the given station over the past "***m=18***" hours.

***Question 6:*** *Pg. 11, line 22-line 28. The description in this paragraph needs further clarification. I believe that you are subtracting the 1-hr forecast made from April 15, 19:00 (so valid at 20:00?) from all the PM10 observations between 8:00 and 19:00 for assimilation. Or is it the one hour forecast valid at each specific time of the observation? In which case does each observation from each station require different 18hour input? Why is the 1-hr forecast chosen rather than the actual value coming from the machine-learning method? How is a forecast made from a machine learning method? Similarly, is the 12-hr forecast from April 15, 19:00 (so valid at April 16, 07:00) added to all the forecast dust PM10 values to compare to the observations?*

***Reply****:* Thanks for pointing out this issue, the original context is indeed ambiguous.

Regarding the 0h forecast: Actually 1h forecast is valid at each specific time of the observation, e.g., forecast of 20:00 is valid at 19:00. In addition, there is no need to use 1h forecast as the bias level, 0h forecast (simulation of 20:00 is also valid at that moment) could make our system work and give a better performance. We have conducted the non-dust bias simulation again but now with "t=0", as well as the corresponding assimilation test. Indeed, both of the bias-correction as well as the posterior dust forecast are somewhat improved when using t=0. New results can be mainly found in Fig.3b (non-dust simulation vs. PM10); ***Fig.4d-4e*** (spatial distribution of the non-dust level and bias corrected dust observations); ***Fig.5*** (time series in 6 cities); ***Fig.7d*** (posterior emission field by assimilating bias-corrected data using LSTM); ***Fig.8d*** (posterior of dust simulation using the bias-corrected measurement by LSTM); ***Fig.9*** (posterior dust forecast in 3 cities) and ***Fig.10*** (RMSE). From the perspective of RMSE, t=0 provides a slightly better result.

Regarding the 12 hours bias forecast: It means the non-dust PM10 forecast of April 16, 07:00 is available at April 15, 19:00. To make these 0h and 12h terms clear, we added more explanation on page 12, line 32; on page 13, line 1-2, "*Note that here t=0 forecasts denote the forecasts valid at each specific snapshot of the observations, while the 12 hour forecasts are the forecasts valid 12 hours in advance, e.g., the non-dust PM10 forecast at April 16 07:00 are valid at April 15 19:00.*"

*Question 7:* Pg. 9-11, Section 3.2. How would such a machine learning algorithm be used in practice? Would it need the constant evaluation of running 18hrs of data? Or if a dust storm was forecast, would the calculations be switched on. At what point would the PM10 observations be assimilated. Or is this envisaged mode for a reanalysis style product of PM10 observations?

*Reply*: It is indeed important to clarify how the machine learning for bias simulation can be used in practice. We added new remarks on page 12, line 13-15 "*Our two bias models, LOTOS-EUROS/non-dust and LSTM, could both be used for air quality forecast operationally when there is no dust storm. Once a dust storm is obsereved, the dust emission inversion system will be enabled, the two non-dust PM10 models will then be used in dust observation bias correction.*"

Regarding when to start the dust storm data assimilation: The emission inversion method we used here is 4DVar, we perform the assimilation test at April 15, 19:00. The assimilation window sets are the same as the ones in Jin et al., 2018. More details are given in *Fig. 2* on page 9.

*Question 8:* Pg. 12, line 9-10. You state that the according to the LOTOS-EUROS bias corrected observations, the dust storm seems to have already reached central China which was in reality not the case? How do you know this? Nothing from this figure provides evidence of what the reality was? It would be worth referencing how you know this to be true.

*Reply*: The LSTM bias estimation is validated to give the best performance in general. The bias-corrected dust observations by LSTM (in Fig 4.e1 on page 15) indicate that there is no dust in the central China, we consider it is more close to the reality than Lotos-Euros bias corrected observations in Fig 4.c1. However, we agree with the reviewer that it is incorrected (100% sure) to say "*in reality*". So, "*was in reality not the case*" is now changed to "*was probably not the case*" on page 14, line 14-15.

*Question 9:* Pg. 13, line 17-19. This to me provides evidence that the LOTOS-EUROS model is not doing a good job of predicting the non-dust PM10 and so is never going to match the performance of the machine learning algorithm. Is there a reason for choosing this model or not exploring CTMs that may provide a better match?

*Reply*: As all CTM's, also Lotos-Euros is under permanent development. Its current performance with respect to aerosols is similar to other CTM models in the Marcopolo-Panda project (Brasseur et al., 2019; Petersen et al., 2019). The ultimate goal is a very accurate simulation. In the current state of development, the machine learning algorithm is doing a better job in the measurement sites as a computationally efficient method. However, machine learning can give prediction at the measurement sites while the CTM can provide simulation results over the entire 3D fields. Machine learning could therefore never fully replace a CTM, and when both are available, it makes sense to compare their ability for use in a bias correction.

The errors in forecast-observation are mainly attributed to inaccuracy in weather forecast and errors in the adopted surface emission schemes. There is definitely space to improve it using nudging techniques like data assimilation. However, it takes lots of efforts and is not exploited in this study yet. To explain this, we add new remarks on page 10, line 3-9 "*The operational CTM Lotos-Euros over the China is in its early phase of development as well as the other six CTMs used in the MarcoPolo-Panda project.*"

*The purpose of the project is to diagnose statistical differences between the ensemble model simulations and observations. An important objective is to determine ways by which the models could be improved. These differences are mostly attributed to inaccuracy in the weather forecast and errors in the adopted surface emissions (Brasseur et al. 2019, Petersen et al. 2019). Indeed, there is room for minimizing the forecast-observation differences using nudging methods like data assimilation, which takes considerable efforts and not yet exploited in this study.*"

Also in the Section of **Future work** on page 25, line 2-8, we mentioned that "***Both our CTM and machine learning based bias correction methods have room for improvements. It might be useful to improve the CTM simulations by assimilating PM10 observations during the hours where no dust storms are present, and use these improved simulations to remove the non-dust part of the observations during an event. These additional assimilations would then involve repeated forward ensemble bias-model runs which could be computationally expensive.***"

**Question 10:** *Pg. 18, line 20. Again, you state that the LOTOS-EUROS observations over-estimate the observations and yet I can't see anything in Figure 8 that shows the observations, so that we know this.*

**Reply**: "***they still overestimate the dust observations***" is now changed to "***they are still likely to overestimate the real dust levels***" on page 20, line 9-10.

**Question:** *Pg. 18, line 31-33. It is interesting to me that although the peak is a better match with the machine learning algorithm, the forecast is actually slightly better with the LOTOSEUROS model. Do you have any feeling for why this may be?*

**Reply**: It could be explained by the context on page 20, line 24-29 "***This can be explained from the fact that the dust storm is a strong flow-dependent phenomenon in which concentrations at a certain location are strongly correlated to earlier concentrations at upwind locations. For Hohhot, only a limited number of observation sites is located upwind, and therefore hardly any data is available to constrain the concentrations at this location. To improve the forecast at Hohhot it will be necessary to have additional observation data, for example from sites actually within the source region, or from satellites observing the aerosol load over the source region (Jin et al. 2019)***".

*Typos:*

*Pg. 1, line 13. I think this should read 'The latter is trained by learning using two years of historical samples'.*
**Reply**: Accepted

*Pg. 2, line 9. Should be 'progress has also been made'*
**Reply**: Accepted

*Pg. 2, line 19. Should be 'A wide variety of data assimilation techniques have been used...'*
**Reply**: Accepted

*Pg. 2, line 28. Should be 'In the presence of biases...'*
**Reply**: Accepted

*Pg. 10, line 25. Should be 'Earlier studies showed that the...'*
**Reply**: Accepted

[revised manuscript text omitted]

---

## Author Comment (AC3) · 12 Jul 2019

**Response to Referee #3:** We would like to thank the referee for the careful review and the insightful comments, which helps us to further improve the quality of the manuscript.

Our response follows (*the reviewer's comments are in italics*)

*General comments:*
*The aim of this work is to develop a methodology able to use PM10 as a tracer of dust with the last end of assimilating PM10 to improve the forecast of dust storms. The deterrent of direct assimilation of PM10 is that it does not only encompass dust but also other aerosol flavors that are more significant where anthropogenic sources of aerosols are present. To isolate dust and non-dust aerosols the idea is to consider non-dust as bias in the PM10 records. Then the problem is reduced to apply bias correction methods to PM10. Here the authors propose to use machine learning and chemical transport model for bias correction. Results show a more accurate forecast of dust storm if the PM10 assimilated is isolated form non-dust aerosols using the machine learning method.*

*The paper is well written, structured and results support the conclusion archived. The topic is innovative since it is one of the first works to apply machine learning to bias correction to isolate non-dust aerosols from PM10. In addition, the methodology developed here goes beyond to improve the forecast of dust storm and it can benefit the assimilation of other magnitudes. This study match with the ACP topics and its quality is close to its standards.*

*However, there are some general comments that need to be addressed or clarified along with the manuscript.*

***Question:*** *1) How this approach guarantees the "bias" in PM10 that you are correcting is non-dust and it is not a "real" bias present in your PM10 records?*
***Reply***: We agree with the reviewer that the PM10 observation itself might have "real" (or "native") bias. However, it is unknown, nor can be quantified using any method as we know. We hence assume that the PM10 measurements are unbiased, but they are biased when they are used as proxy as dust observations. To point out the possibility of "real" bias in PM10, we add some remarks on page 8, line 2-4 "***Note that the PM10 measurements themselves might also contain 'native' biases due to the incorrect sensor reading or systematic errors. However, this part of the bias in the PM10 observations is unknown and not considered in this study.***"

***Question:*** *2) How do you know a better performance of LSTM for bias correction (or dust isolation from PM10) and also better dust forecast if you are not comparing against dust records? I am aware of the lack of availability of them but at least it will be necessary to actually know that your PM10 without bias is dust by comparing with some station or a satellite image.*
***Reply***: Yes, it will be better if we would have "pure" dust observations with which we could validate our dust forecasts. However, almost all the observations available measure the sum of dust and non-dust aerosols, e.g., PM10 or aerosol optical depth. To fully evaluate our system, we validate the bias forecast as well as the dust storm forecasts using the bias-corrected data.
The bias simulation is actually the operational full aerosol forecast when there is no dust storm. We compare it to the real PM10 observations in test period (April and May) in ***Fig.3*** on page 13. Since the test period includes the dust storm event (April 14 to 16), we now plot the scatters using different marker for dust and non-dust period. The new ***Fig.3b*** and ***Fig.3c*** show that our method can accurately simulate

the non-dust PM10. The bias correction performance can also be seen in the time series at 6 different cities in ***Fig.5*** on page 16. The comparison indicates that our bias simulation is very close to the non-dust aerosol level hence can be used to calculate the "real" dust level.

In addition, since we only have PM10 measurements and don't have the pure dust observation during the dust storm, to evaluate the final dust storm forecast, we calculate the difference between full aerosol simulation (dust forecast + non-dust forecast) and PM10 measurement. The RMSE in ***Fig.10*** on page 24 also indicates our forecast has a good and constant performance in the full aerosol modeling during the dust storm.

***Question:*** *3) Also, the main caveat of the machine learning methods is the availability of a long series of records and the computational time. Do you think that this method will be able to correct the PM10 bias obtained a few hours (e.g. 12 h) before a dust storm and be ready to assimilate and perform the forecast a few hours later? The real applicability of LSTM is not clear. This point needs a little more discussion.*

***Reply***: We have machine learning based non-dust aerosol forecast in this study, they are 0 and 12 hours in advance, respectively. The former one is used for bias correction, hence the bias simulation only need be ready at the specific moment that we want to have dust storm data assimilation. When the dust forecast is ready after the assimilation analysis, the later one (12 h) will be added to the posterior dust storm forecast and then treated as the full aerosol forecast. To clarify this, we added extra remarks on page 12, line 31-32, on page 13, line 1-2 by saying "***When we perform the assimilation analysis at April 15, 19:00, the short period of t=0 h forecast will be treated as the non-dust levels in the bias correction of the original PM10 measurements. Note that here t=0 forecasts denote the forecasts valid at each specific snapshot of the observations, while the 12 hours forecasts are valid 12 hours in advance, e.g., the non-dust PM10 forecast (12 h) at April 16 07:00 is valid at April 15 19:00.***"

We also add more remarks to explain the computational efficiency regarding to our machine learning work on page 12, line 3-5 "***The machine learning model for non-dust PM10 forecast is trained site by site, with the hyper-parameters shown in Table.1. With the following hyper-parameters, the machine learning model training costs several minutes for each site. The model training in each site is independent hence the whole workload is highly parallelizable.***"

***Question:*** *4) Finally, the conclusion about what method is better for bias correction looks like that is the same method that better "simulate" the non-dust aerosols. Then, the worst bias correction with LOTOS-EUROS model is just the worst performance in reproducing non-dust aerosols. This simplification is not straightforward needs to be clarified.*

***Reply****:* We assume that the PM10 measurements themselves are not biased, but if we use them as proxy of dust observations, then they are biased. The real dust level is calculated by subtracting the non-dust level from the PM10 observations. Therefore, we make the conclusion that the method that better simulate the non-dust aerosol is the better bias correction method.

To make the conclusion more justified, we added more statistical descriptions on page 24, line 1-5 "***The two methods to estimate the non-dust part of the PM10 load have been validated. The simulations by the LOTOS-EUROS/non-dust model in general underestimate the PM10 concentrations. The root mean square error stays at a relative high level of 89.4 ug/m3. It is mainly caused by missing emissions and aerosol components such as secondary organic matter. In comparison, the data-driven machine learning model agrees more closely with the real measurements, the RMSE declines to 58.6 ug/m3.***"

Also on page 24, line 14-19 "*The dust emissions estimated using the assimilation can be used to drive a dust forecast. When the original PM10 observations were used in the assimilation, the forecast skill of the system actually decreased due to the strong overestimation of dust concentrations, the RMSE rose from averagely 230 (prior forecast) to 300 ug/m3. Better forecasts are obtained when using the model-based and especially the machine learning based bias-corrected observations. The RMSE of the former one was reduced to 200 ug/m3 while the RMSE of the latter one further declined to 150 ug/m3.*"

*Minor comments*
*P4-L29. to large -> too large*
**Reply**: "*to large extent*" is changed to "*to a large extent*"

*P5-L2. Can easily applied -> can easily be applied*
**Reply**: Accepted

*P7-L3. Following Jin et al., 2018*
**Reply**: Accepted

*P9-L15. Anthropogenic emissions are from MEIC but it is not clear where natural emissions are obtained. Are these natural emissions; sea salt, biogenic emissions, and wildfires, but not dust?*
**Reply**: To clarify the source of the nature emission, we added new remarks on page 9, line 16-17 "*Natural emissions included are the sea salts that are calculated online, biogenic emissions that are calculated online using the MEGAN model (Guenther et al., 2018), and wild fires which were taken from the operational GRAS product (Kaiser et al. 2012).*"

*P10-L11 How LSTM is used needs little more details. It is not clear why these elements of the input vector have been chosen. Do these observations correspond to blue dots in Fig. 1? Why do you use nearby sites for PM2.5 but not for the other species? Is the training period only past 18 h or from January 2013 to March 2015 as it is said in P11-L4? Are these data available hourly? P11-L23 What criteria do you use to consider that a series has high data missing?*
**Reply**: Yes, the LSTM indeed needs more clarifications.
Regarding to "*why these elements of inputs are selected*": We added new explanation on page 11, line 28-31 "*Statistical analysis tests are conducted which not only indicate a strong correlation between the non-dust PM10 and air quality measurements in the give sites, but also show that the predictor (non-dust PM10) is correlated to the observation indices (especially the PM2.5) at nearby sites.*"
Regarding to site map in Fig. 1: We added new remarks on page 6, line 4-5 "*At the present, the monitoring network has grown to 1,500 field stations covering all over China as shown in Fig. 1*"
Regarding to training period: Remarks are added on page 11, line 4-6 "*The training dataset covers the period from January 2013 to March 2015. In other words, the LSTM model L is trained to best fit the samples from this period. The two months April and May 2015 in which the studied dust event occurred is set as the testing period.*"
Regarding to "*Are these data available hourly*": Yes, all the measurements are the hourly averages. We added new remarks to clarify this on page 10, line 25 "*hourly observations of PM2.5, SO2, NO2, O3 and CO from the ……*"
Regarding to the inputs of PM2.5 at nearby sites, data missing rate: We added more remarks on page 11, line 20-34 "*For the non-dust PM10 machine learning forecasts in a given site, observations from its*

*nearby sites are also vital and are used in two ways. First, missing data records are unavoidable in an air quality monitoring network, while the LSTM model training requires an uninterrupted time series of features. In this study, data interpolations of air quality measurements (PM10, PM2.5, SO2, NO2, O3 and CO) are performed using both a linear interpolation and a k-Nearest-Neighbor algorithm (Zhang, 2012) if a site has no more than 30% of missing data. Otherwise, all the measurements in the given sites are abandoned. Generally, more information available from the nearby sites will result in a more accurate interpolation. Second, learning in the presence of data errors is pervasive in machine learning, and the measurements from nearby stations are used to limit their influence. Data errors occur due to incorrect sensor readings, software bugs in the data processing pipeline, or even the inaccurate data interpolation. Statistical analysis tests have been conducted which did not only indicate a strong correlation between the non-dust PM10 and air quality measurements in the given sites, but also show that the predictor (non-dust PM10) is correlated to the observation indices (especially the PM2.5) at its nearby sites. In order to eliminating errors caused by incorrect inputs at the modeling site, the measurements at the nearby stations are considered as the essential indices. In this study, a data instance will only be selected for training the LSTM model if there is at least one nearby site within an empirical radius 0.8\degree (approx 80 km), and a maximum of 3 nearby sites will be randomly selected where observation stations are densely distributed. To save the computation costs on machine learning model training, only the PM2.5 from the nearby sites are included as one of the inputs in this study.*"

*P11-L4 I would like to see (or at least discuss) the sensitivity of the results to the training period. For instance, in the manuscript it is argued that from 01-2013 to 03-2015 the frequency of dust storm was low then it could be considered that all PM10 were nondust. However, yb will be significantly larger during the days with a dust storm and this could affect the regression model. Have you checked if the regression model changes if you remove those days with dust storm?*

**Reply**: Yes, the dust records in the training period would indeed influence the machine learning model. However, the dust storm event we studied is reported to be the most severe one since 2002. Such a large-scale dust events has not been recorded in the training period (2013 to 2015). It is true that cities close to the deserts might have experienced several dust events with less pollutant levels. However, the learning rate on this single sample is set as 0.0001 in our machine learning algorithm. Therefore, these records would not have too much influence on the final machine learning model, whereas to identify these records and remove them takes a lot of efforts.

To clarify this, we also add some new remarks on page 11, line 7-14 by saying "***Dust storms themselves occur with very low frequency. To our knowledge, the studied dust event is the most severe one since 2002, and there are no such large-scale dust events recorded in our training period. Note that cities that are close to the Gobi and Mongolia deserts might have experienced several small- scale dust events with limited increase of dust concentrations. However, the machine learning tries to find the global best fits for the whole training dataset. The default learning rate, which determines the weights are updated during training, on a simple sample is $10^{-4}$ in our machine learning algorithm. Therefore, the PM10 records $y^b$ are very close to the non-dust PM10 concentrations, and the rare dust event records are not excluded from the training dataset for convenience and for the expected little impact on the training result.***"

*P12-L12 In non-dust PM10 evaluation, how do you know that they are dust in Fig. 3? Is it only based on larger numbers?*

*Reply*: We have modified the Fig.3 on page 13. Different markers are used to represent scatters in the dust period (April 14-16) and the rests in April 01 to May 30. The new plots indicate that most of the scatters in the bottom right corner are from the 3 days of the dust period while few parts are form the rest 57 days of non-dust period.

*P11-L1 How do you know LE/non-dust underestimate non-dust PM10? Is it based in Fig. 3 and lower values of PM10?*

*Reply*: Yes, according to the revised ***Fig.3*** and ***Fig.5***. It is clear that the LE/non-dust model underestimates the non-dust $PM_{10}$.

*P11-L5 To show the better agreement of LSTM than LE/no-dust, could you give some number such as correlations?*

*Reply*: The RMSEs of all the three non-dust simulations vs. measurements are given in the revised Fig.3 on page 13. They are also described when evaluating the non-dust simulation performance on page 12, line 19-27 by saying "***The CTM LOTOS-EUROS/ non-dust in general underestimates the non-dust PM10. The forecast results in a relatively large root mean square error (RMSE) 89.4 ug/m3. This could be explained from the fact not all types of particulate matters, such as secondary organic aerosols, are included in the model, and some aerosol emissions are very difficult to estimate (e.g., wood burning by households). The two LSTM forecasts show on average a good agreement with the observations. The RMSEs of the forecasts by the two machine learning models in the two years of training period are reduced to 55.9 and 60.7 ug/m3, and in the two months of test period (excluding the dust event from April 14 to 16) they also stay at comparable low levels of 58.6 and 60.2 ug/m3.***"

Also on page 24, line14-19 "***The two methods to estimate the non-dust part of the PM10 load have been validated. The simulations by the LOTOS-EUROS/non-dust model in general underestimate the PM10 concentrations. The root mean square error stays at a relative high level of 89.4 ug/m3. It is mainly caused by missing emissions and aerosol components such as secondary organic matter. In comparison, the data-driven machine learning model agrees more closely with the real measurements, the RMSE declines to 58.6 ug/m3.***"

*P13-L25. Is a high level of pollution the reason why these sites are chosen to test the performance of LE/non-dust?*

*Reply*: To clearly explain why we choose these six cities, we added some new remarks on page 14, line 25-27 "***These cities were selected because they all experienced a severe pollution and illustrated the general performance of the LOTOS-EUROS/non-dust and LSTM methods. In addition, each of these cities have at least 4 monitoring sites which assured a high accuracy.***"

*P20-L15 In addition to the orography, a reason why in Beijing the PM10 is underestimated could be related to higher PM10 non-dust values? Did you find any relationship.*

*Reply*: We agree with the reviewer about this point. Explanation is added on page 20, line 34; on page 22, line 1-2 "***This suggests that the simulation model simply is not able to increase the dust concentrations here, for example because of uncertainties in the meteorological data, a removal of dust that is too efficient, or because some local sources of dust are absent (equally, non-dust PM10 levels are underestimated).***"

In the section of the evaluation of the forecast skill, I miss comparing with "real" dust records instead of PM10.

***Reply***: Since we never have the "real" dust records/measurements. What we compare is the difference between the full aerosol simulation (dust + non-dust aerosol simulation) and PM10 observations. It is now pointed out on page 22, line 10-12 by saying "***To evaluate the forecast skill of the assimilation(s), the root mean square error (RMSE) of the reference and three posterior full aerosol simulations (dust forecasts plus non-dust predictions) with respect to the observed PM10 over the whole observation sites has been computed for each hour.***"

*Figures, in general, are clear and the captions properly describe them.*
*Fig.1 It is difficult to distinguish the star symbols which denote the cities location. Also,*
***Reply***: We now use double-size markers for the six cities.

*Is N=1352 number of stations used in LSTM?*
***Reply***: Yes. We also add "***LSTM based non-dust PM10 forecast are performed only in stations of blue dot (N=1351)***" in the caption of Fig.1 on page 6.

*Fig. 3 Use properly notation for float numbers (e.g. 2 10-6 instead of 0.00002).*
***Reply***: Fig. 3 is modified. See new version on page 13.

*Fig. 6 Unify the legends and try it does not overlap with the time series.*
***Reply***: Accepted. Legends in ***Fig.5*** on page 16 are modified.

[revised manuscript text omitted]